# WHEN EMBEDDING MODELS MEET: PROCRUSTES BOUNDS AND APPLICATIONS

## ABSTRACT

Embedding models trained separately on similar data often produce representations that encode stable information but are not directly interchangeable. This lack of interoperability raises challenges in several practical applications, such as model retraining, partial model upgrades, and multimodal search. Driven by these challenges, we study when two sets of embeddings can be aligned by an orthogonal transformation. We show that if pairwise dot products are approximately preserved, then there exists an isometry that closely aligns the two sets, and we provide a tight bound on the alignment error. This insight yields a simple alignment recipe, Procrustes post-processing, that makes two embedding models interoperable while preserving the geometry of each embedding space. Empirically, we demonstrate its effectiveness in three applications: maintaining compatibility across retrainings, combining different models for text retrieval, and improving mixed-modality search, where it achieves state-of-the-art performance.

## 1 INTRODUCTION

Representing objects as dense vectors is central to many key applications of machine learning (Bengio et al., 2013). In recommender systems, low-dimensional embeddings capture user preferences for content (Koren et al., 2009). In text or image search applications, embedding models enable efficient semantic similarity and relevance computation (Deerwester et al., 1990; Reimers & Gurevych, 2019).

Embedding models are typically trained to capture notions of similarity between objects as geometric relationships in Euclidean space. Specifically, loss functions underpinning representation learning methods usually depend only on distances or dot-products between embeddings. Such loss functions are therefore orthogonally invariant: any rotation and reflection of the embedding space yields an identical loss function value. This invariance makes embeddings under-specified. Two distinct models might capture similar geometrical relationships but produce embeddings that are not directly comparable. This becomes problematic when multiple embedding models are used together.

**Model retraining.** To capture concept drift, it is sometimes necessary to retrain the embedding model on fresh data, resulting in successive versions of an embedding space (Shiebler et al., 2018; Steck et al., 2021). Because the spaces are not aligned, downstream systems trained on embeddings from one version cannot be used with embeddings from another version. This creates challenges when embedding models and downstream systems are retrained at different cadences (Hu et al., 2022).

**Partial upgrades.** In retrieval, relevance is often predicted by the dot product between query and document embeddings. A practical difficulty arises when the query model is upgraded but document embeddings cannot be recomputed, either because the raw documents are not available (Morris et al., 2023; Huang et al., 2024), or recomputation is too costly (Shen et al., 2020; Arora et al., 2020).

**Multimodal embeddings.** Models such as CLIP (Radford et al., 2021) and SigLIP (Zhai et al., 2023) embed text and images into a shared space, enabling cross-modal comparison. Yet these models have been observed to exhibit a *modality gap*, where embeddings cluster by modality into distinct regions of Euclidean space (Liang et al., 2022). This prevents meaningful comparison of dot products across heterogeneous pairs of modalities, and degrades the performance of mixed-modality search (Li et al., 2025).

| Misaligned embeddings | Transforming $X$ orthogonally | $\bar{X}$ is aligned with $Y$ |
|---|---|---|
| 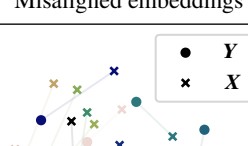 | 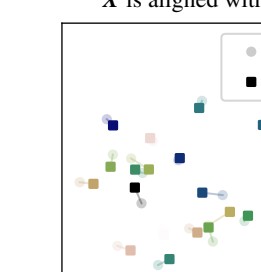 | 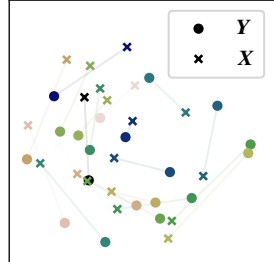 |

Figure 1: We start with two sets of embeddings $X$ and $Y$ that approximately preserve distances but are unaligned (*left*). We find $\bar{X}$ by apply the orthogonal Procrustes transformation to $X$ (*center*). $\bar{X}$ retains the exact geometry of $X$ but the embeddings are now aligned with $Y$ (*right*).

Driven by these practical settings, we consider the problem of aligning two sets of vectors that approximately preserve geometry. Specifically, we study the orthogonal Procrustes problem (Hurley & Cattell, 1962; Schönemann, 1966), which asks for an orthogonal transformation that minimizes the average squared distance between corresponding vectors in each set. In this paper, we ask the question: How well does the optimal orthogonal transformation align the two sets of vectors, assuming only that dot products are approximately preserved across the two sets? In Section 3, we address this question by providing a tight bound on the average distance between a vector from the first set and the aligned version of the corresponding vector in the second set. In the regime of interest, our bound improves on the state of the art (Tu et al., 2016; Arias-Castro et al., 2020; Pumir et al., 2021), and settles a conjecture of Arias-Castro et al. (2020, Remark 1).

These results suggest a simple recipe to make two embedding models interoperable: Post-process embeddings produced by one model by applying the orthogonal Procrustes transformation with respect to the other model. This maximizes cross-model alignment without affecting the geometry of the embeddings produced by each model. We illustrate this procedure in Figure 1. In Section 4, we empirically evaluate the effectiveness of Procrustes post-processing across the three applications introduced above. We find that it successfully addresses the corresponding challenges, without any modification to the underlying representation learning method. Among others, we find that *a*) post-processing successive model versions effectively solves the version mismatch problem, *b*) using a more powerful query embedding can dramatically improve text retrieval performance, but only once it is aligned with the document embedding model, and *c*) Procrustes post-processing provides state-of-the art performance on a mixed-modality search benchmark, outperforming recent work by Li et al. (2025).

**Contributions.** Our main contribution is a theoretical result establishing that if two embedding models approximately preserve dot products, they can be aligned through an orthogonal transformation, enabling interoperability. While orthogonal alignment is a well-established technique and is already used in practice, we believe its theoretical underpinnings and broad applicability remain underappreciated. To this end, we complement our analysis with experiments in three real-world applications, both reinforcing prior empirical findings and providing new insights.

## 1.1 Preliminaries and notation

We consider two sets of $N$ vectors in $\mathbf{R}^D$, arranged into $D \times N$ source and target embedding matrices $X = \begin{bmatrix} x_1 & \cdots & x_N \end{bmatrix}$ and $Y = \begin{bmatrix} y_1 & \cdots & y_N \end{bmatrix}$, respectively. We assume that the $i$th vector encodes the same object across both embeddings. For example, $x_i$ and $y_i$ consist of the same text passed through two different text embedding models, or they represent the same user in a recommender system application. Given a function $f : \mathbf{R}^D \to \mathbf{R}$, we denote its empirical average over the embeddings as $\mathbf{E}_i[f(x_i)] \doteq (1/N) \sum_{i=1}^{N} f(x_i)$.

We say that the $D \times D$ matrix $Q$ is orthogonal if $Q^\top Q = I_D$, where $I_D$ is the identity matrix, and we denote the set of $D$-dimensional orthogonal matrices by $\mathcal{O}_D$. Orthogonal transformations are

---

**Algorithm 1** Orthogonal Procrustes (Schönemann, 1966)

---

**Require:** $\boldsymbol{X}, \boldsymbol{Y} \in \mathbf{R}^{D \times N}$
**Ensure:** $\boldsymbol{Q}^\star \in \arg\min_{\boldsymbol{Q}} \|\boldsymbol{Q}\boldsymbol{X} - \boldsymbol{Y}\|_F$ subject to $\boldsymbol{Q}^\top \boldsymbol{Q} = \boldsymbol{I}$
 1: $\boldsymbol{U}\boldsymbol{\Sigma}\boldsymbol{V}^\top \leftarrow$ singular value decomposition of $\boldsymbol{Y}\boldsymbol{X}^\top$
 2: $\boldsymbol{Q}^\star \leftarrow \boldsymbol{U}\boldsymbol{V}^\top$

---

isometries, i.e., they preserve distances and dot products exactly.[1] We would like to find an orthogonal transformation $\boldsymbol{Q} \in \mathcal{O}_D$ such that $\bar{\boldsymbol{x}}_i \doteq \boldsymbol{Q}\boldsymbol{x}_i$ for all $i \in [N]$ and $\|\boldsymbol{y}_i - \bar{\boldsymbol{x}}_i\|_2$ is small, on average. Intuitively, we think of $\boldsymbol{Q}$ as aligning $\boldsymbol{X}$ and $\boldsymbol{Y}$. Formally, we seek to solve

$$\boldsymbol{Q}^\star \in \arg\min_{\boldsymbol{Q} \in \mathcal{O}} \|\boldsymbol{Q}\boldsymbol{X} - \boldsymbol{Y}\|_F, \tag{1}$$

where $\|\cdot\|_F$ is the Frobenius norm. This is known as the orthogonal Procrustes problem (Hurley & Cattell, 1962), and $\|\boldsymbol{Q}^\star \boldsymbol{X} - \boldsymbol{Y}\|_F$ is referred to as the Procrustes distance between $\boldsymbol{X}$ and $\boldsymbol{Y}$. In a seminal paper, Schönemann (1966) introduces a simple, computationally-efficient procedure for solving (1), which we describe in Algorithm 1.

## 2 RELATED WORK

**Isometries and approximate isometries.** If $\boldsymbol{X}$ and $\boldsymbol{Y}$ approximately preserve geometry, we can view the mapping $\boldsymbol{x}_i \mapsto \boldsymbol{y}_i$ through the lens of *approximate isometries*. The Mazur-Ulam theorem states that every exact isometry in Euclidean space is an affine transformation (Mazur & Ulam, 1932). Building on this, Hyers & Ulam (1945) show that mappings that preserves distances approximately can be well-approximated by exact isometries, but their result applies only to mappings that are defined on entire vector spaces, e.g., all of $\mathbf{R}^D$. Fickett (1982) and Alestalo et al. (2001) study extensions of this result to bounded subsets of $\mathbf{R}^D$, but the resulting guarantees are impractical.

**Theory of orthogonal Procrustes.** Söderkvist (1993) derives a perturbation bound for orthogonal Procrustes in the special case where the alignment is restricted to rotations (orthogonal matrices with positive determinant). Tu et al. (2016) introduce the first practical bound on the Procrustes distance in terms of the distance between Gram matrices, later refined by Pumir et al. (2021). Arias-Castro et al. (2020) independently obtain a similar result and study applications to multi-dimensional scaling. As we discuss in Section 3, our bounds are significantly tighter in the regime of interest. Recently, Harvey et al. (2024) relate several representational similarity measures, including the Procrustes distance, and develop a result similar to ours but restricted to centered embedding matrices.

**Applications of embedding alignment.** Shiebler et al. (2018) and Steck et al. (2021) discuss practical challenges of embedding models in large-scale online services. Both highlight the need for periodic retraining to combat concept drift and difficulties created by misaligned successive versions, including organizational challenges. To address these, El-Kishky et al. (2022), Hu et al. (2022), and Gan et al. (2023) propose modifications to training procedures to produce aligned embeddings for recommender systems. A different line of work studies embedding alignment for visual search, aiming to avoid costly backfilling (recomputing embeddings for existing images under a new model). Shen et al. (2020) and Meng et al. (2021) introduce training objectives that promote compatibility across successive model versions.

**Embedding alignment with orthogonal Procrustes.** Singer et al. (2019) and Tagowski et al. (2021) apply orthogonal Procrustes to align successive node embeddings in time-varying graphs, demonstrating effectiveness for node classification and link prediction. In natural language processing, alignment methods are widely used to relate word embeddings across languages. Early work employs unconstrained linear transformations (Mikolov et al., 2013), but subsequent papers (Xing et al., 2015; Artetxe et al., 2016) show the importance of preserving each language's embedding geometry. Grave et al. (2019) address a harder problem where no dictionary is available, requiring joint optimization of word mapping and embedding alignment. For a comprehensive overview, we refer the reader to Ruder et al. (2019). In recommender systems, concurrent work by Zielnicki & Hsiao (2025) explores

---

[1]For example, it is easy to verify that for any $\boldsymbol{Q} \in \mathcal{O}_D$ and any $\boldsymbol{u}, \boldsymbol{v} \in \mathbf{R}^D$, we have $(\boldsymbol{Q}\boldsymbol{u})^\top \boldsymbol{Q}\boldsymbol{v} = \boldsymbol{u}^\top \boldsymbol{v}$.

orthogonal Procrustes for aligning successive embedding model versions, closely related to our study in Section 4.1.

# 3 UPPER BOUND ON THE PROCRUSTES DISTANCE

Our motivating applications require combining two embedding models that encode similar geometric relationships but are not directly aligned. This raises the question: Under what conditions can two embedding matrices be well-aligned by an orthogonal transformation? We answer this question by providing a tight upper bound on the Procrustes distance, assuming only that pairwise dot products are approximately preserved across the two sets of vectors.

**Theorem 1.** *Let* $\boldsymbol{X}, \boldsymbol{Y} \in \mathbf{R}^{D \times N}$, *and let* $\varepsilon = \|\boldsymbol{X}^\top \boldsymbol{X} - \boldsymbol{Y}^\top \boldsymbol{Y}\|_F$. *Then,*

$$\min_{\boldsymbol{Q} \in \mathcal{O}_D} \|\boldsymbol{Q}\boldsymbol{X} - \boldsymbol{Y}\|_F \leqslant (2D)^{1/4}\sqrt{\varepsilon}.$$

*Proof (sketch).* The key idea is to identify a suitable canonical factorization of the Gram matrix $\boldsymbol{X}^\top \boldsymbol{X}$. We find that the matrix absolute value $|\boldsymbol{X}| \doteq (\boldsymbol{X}^\top \boldsymbol{X})^{1/2}$ provides the appropriate notion. An extension of the Powers-Størmers inequality (Kittaneh, 1986) allows us to bound $\||\boldsymbol{X}| - |\boldsymbol{Y}|\|_F^2$ as a function of $\|\boldsymbol{X}^\top \boldsymbol{X} - \boldsymbol{Y}^\top \boldsymbol{Y}\|_F$. With some more work, we show how to bound $\min_{\boldsymbol{Q} \in \mathcal{O}_D} \|\boldsymbol{Q}\boldsymbol{X} - \boldsymbol{Y}\|_F$ as a function of $\||\boldsymbol{X}| - |\boldsymbol{Y}|\|_F$. The full proof is provided in Appendix A.1. $\square$

Intuitively, the condition $\|\boldsymbol{X}^\top \boldsymbol{X} - \boldsymbol{Y}^\top \boldsymbol{Y}\|_F \leqslant \varepsilon$ measures how closely dot products are preserved across $\boldsymbol{X}$ and $\boldsymbol{Y}$. Theorem 1 shows that this stability of dot products translates directly into stability under alignment: the optimal orthogonal transformation mapping $\boldsymbol{X}$ close to $\boldsymbol{Y}$ has alignment error at most $O(\sqrt{\varepsilon})$. In particular, small deviations in dot products guarantee small distances between corresponding vectors once they are aligned. The dependence of Theorem 1 on both $D$ and $\varepsilon$ is tight, and in Appendix A.2 we provide an explicit example that achieves equality. The next corollary reformulates the theorem in terms of the average squared error in dot products, providing a measure of stability that is easier to interpret.

**Corollary 1.** *Let* $\boldsymbol{X}, \boldsymbol{Y} \in \mathbf{R}^{D \times N}$, *and let* $\delta^2 = \mathbf{E}_{i,j}\left[(\boldsymbol{x}_i^\top \boldsymbol{x}_j - \boldsymbol{y}_i^\top \boldsymbol{y}_j)^2\right]$. *Let* $\boldsymbol{Q}^\star$ *be the output of Algorithm 1, and denote by* $\bar{\boldsymbol{x}}_i \doteq \boldsymbol{Q}^\star \boldsymbol{x}_i$ *the embedding aligned with* $\boldsymbol{y}_i$. *Then,*

$$\mathbf{E}_i\left[\|\bar{\boldsymbol{x}}_i - \boldsymbol{y}_i\|^2\right] \leqslant \sqrt{2D}\delta.$$

*Proof.* Setting $\varepsilon = N\delta$ in Theorem 1 and using the definition of the Frobenius norm gives the result, since $\boldsymbol{Q}^\star \in \arg\min_{\boldsymbol{Q} \in \mathcal{O}_D} \|\boldsymbol{Q}\boldsymbol{X} - \boldsymbol{Y}\|_F$. $\square$

Finally, an important special case arises when embeddings are normalized, i.e., $\|\boldsymbol{x}_i\| = \|\boldsymbol{y}_i\| = 1$, so that dot products coincide with cosine similarities. In this setting, we can also bound the deviation of cross-similarities $\bar{\boldsymbol{x}}_i^\top \boldsymbol{y}_j$ with respect to both $\boldsymbol{y}_i^\top \boldsymbol{y}_j$ and $\boldsymbol{x}_i^\top \boldsymbol{x}_j$.

**Corollary 2.** *Let* $\boldsymbol{X}, \boldsymbol{Y} \in \mathbf{R}^{D \times N}$ *be embedding matrices with unit-norm columns, and let* $\delta^2 = \mathbf{E}_{i,j}\left[(\boldsymbol{x}_i^\top \boldsymbol{x}_j - \boldsymbol{y}_i^\top \boldsymbol{y}_j)^2\right]$. *Let* $\boldsymbol{Q}^\star$ *be the output of Algorithm 1, and denote by* $\bar{\boldsymbol{x}}_i \doteq \boldsymbol{Q}^\star \boldsymbol{x}_i$ *the embedding aligned with* $y_i$. *Then,*

$$\mathbf{E}_{i,j}\left[\|\bar{\boldsymbol{x}}_i^\top \boldsymbol{y}_j - \boldsymbol{y}_i^\top \boldsymbol{y}_j\|^2\right] \leqslant \sqrt{2D}\delta, \qquad \mathbf{E}_{i,j}\left[\|\bar{\boldsymbol{x}}_i^\top \boldsymbol{y}_j - \boldsymbol{x}_i^\top \boldsymbol{x}_j\|^2\right] \leqslant \sqrt{2D}\delta.$$

*Proof.* For the first result, we have

$$\mathbf{E}_{i,j}\left[\|\bar{\boldsymbol{x}}_i^\top \boldsymbol{y}_j - \boldsymbol{y}_i^\top \boldsymbol{y}_j\|^2\right] = (1/N^2)\|(\boldsymbol{Q}^\star \boldsymbol{X})^\top \boldsymbol{Y} - \boldsymbol{Y}^\top \boldsymbol{Y}\|_F^2 = (1/N^2)\|(\boldsymbol{Q}^\star \boldsymbol{X} - \boldsymbol{Y})^\top \boldsymbol{Y}\|_F^2$$
$$\leqslant (1/N^2)\|\boldsymbol{Q}^\star \boldsymbol{X} - \boldsymbol{Y}\|_F^2 \|\boldsymbol{Y}\|_F^2 \leqslant \sqrt{2D}\delta,$$

where the first inequality follows from the Cauchy-Schwarz inequality, and the second inequality follows from Corollary 1 and from the fact that, since $\boldsymbol{Y}$ has unit-norm columns, $\|\boldsymbol{Y}\|_F^2 = N$. The second result follows from the first by exchanging $\boldsymbol{X}$ and $\boldsymbol{Y}$ and noticing that $\bar{\boldsymbol{x}}_i^\top \boldsymbol{y}_j = \boldsymbol{x}_i^\top \bar{\boldsymbol{y}}_j$. $\square$

**Comparison to prior work.** We briefly contrast our result with those of Tu et al. (2016) and Arias-Castro et al. (2020). Under the additional assumption that $\boldsymbol{X}$ has full rank, they bound $\min_{\boldsymbol{Q} \in \mathcal{O}_D} \|\boldsymbol{Q}\boldsymbol{X} - \boldsymbol{Y}\|_F$ by $\sigma_{\min}^{-1}\varepsilon$ (up to a constant factor), where $\sigma_{\min}$ is the smallest singular value of $\boldsymbol{X}$. In contrast to theirs, our bound is entirely data-independent. Moreover, the setting most relevant to our applications is $\varepsilon = N\delta$ with $\delta$ fixed and small but $N$ large, in which case typically $\sqrt{\varepsilon} \ll \varepsilon$ and our bound is tighter. This is highlighted in the framing of Corollary 1 where our bound is independent of $N$, whereas their bound scales as $O(N\delta^2)$. In a different line of work, Harvey et al. (2024) prove a bound similar to ours. Their result, however, applies only to centered embedding matrices ($\mathbf{E}_i[\boldsymbol{x}_i] = \mathbf{E}_i[\boldsymbol{y}_i] = \mathbf{0}$). By contrast, our bound does not required centered embeddings.

## 4 EXPERIMENTAL EVALUATION

In this section, we take an empirical perspective. We investigate the effectiveness of orthogonal Procrustes across three practical applications where distinct embedding models need to be aligned: Model retraining (Sec. 4.1), partial upgrades (Sec. 4.2), and mixed-modality search (Sec. 4.3).

### 4.1 MAINTAINING COMPATIBILITY ACROSS RETRAININGS

In some representation learning applications, it is standard practice to periodically retrain embedding models on fresh data in order to capture concept drift, i.e., evolving relationships between objects. To study this setting, we consider the MovieLens-25M dataset, which consists of 25M movie ratings and associated genre metadata from an online recommender system (Harper & Konstan, 2015). We train low-dimensional user and item embeddings using a BPR matrix factorization model that predicts positive movie ratings (Rendle et al., 2009). Details of training and hyperparameter selection are provided in Appendix B.1. Successive model versions are obtained by training on data consisting of all ratings in the six months preceding a given month $t$, for 4 consecutive months. This setup mirrors a realistic scenario in which production recommender systems are retrained on a regular cadence.

Matrix factorization models are invariant to orthogonal transformations. Consequently, successive versions of the embeddings are misaligned by default, which poses challenges for downstream systems that consume embeddings as input. Such systems must either be retrained synchronously with the embedding model (a stringent and often impractical requirement) or the embeddings must be made interoperable across versions.

Orthogonal Procrustes post-processing provides a simple and attractive solution to this problem (Zielnicki & Hsiao, 2025). By aligning embeddings from version $t$ to those of a fixed reference version $t_0$, we obtain interoperability across retrainings without modifying the training objective or distorting the geometry of individual embedding spaces. We compare this approach against several alternatives.

**Warmstart.** Initialize embeddings of version $t$ with those of version $t_0$.

**Autoencoding loss.** Add a regularization term penalizing squared distances between the embeddings of version $t$ and $t_0$ (El-Kishky et al., 2022).

**BC-Aligner.** The method of Hu et al. (2022), which jointly learns embeddings and a linear transformation aligning embeddings from $t$ to $t_0$ during training.

**Linear.** Post-process the embeddings with the best-fitting linear transformation. Relaxing the orthogonality constraint allows improved alignment but sacrifices geometry preservation.

All of the competing methods introduce inductive biases, either through modifications to the loss function or by altering the geometry of the embedding space after training. Orthogonal Procrustes is unique that it does not introduce any additional inductive biases.

**Similar movies retrieval.** We first evaluate alignment methods on a similar-movie retrieval task. We select the 5000 movies with the most ratings. For each movie $i$ in an embedding space $\boldsymbol{X}$ corresponding to $t > t_0$, we rank all other movies by decreasing dot product $\boldsymbol{x}_i^\top \boldsymbol{x}_j$ and record the top-100 most similar movies. Given the reference embeddings $\boldsymbol{Y}$ from $t_0$, and aligned embeddings $\bar{\boldsymbol{X}}$ from $t$, we approximate similarity as $\bar{\boldsymbol{x}}_i^\top \boldsymbol{y}_j$ and report recall@100. Figure 2 (*left*) shows the results. As expected, unaligned embeddings fail to recover similar movies. Orthogonal Procrustes

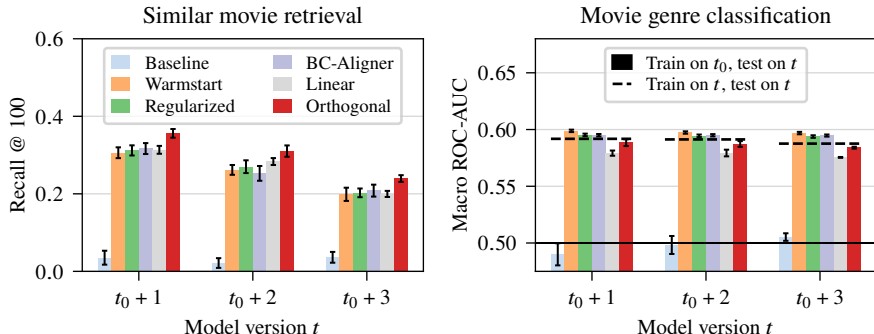

Figure 2: Retraining experiments on the MovieLens dataset. Models trained on embeddings from version $t_0$ are combined with embeddings from version $t > t_0$.

achieves the best performance among alignment methods, likely owing to the fact that $\bar{X}$ preserves the geometry of $X$ exactly.

**Movie genre prediction.** We also evaluate a downstream classification task: predicting the genres of a movie from its embedding. To this end, we partition the movies into training and test sets. For each genre, we train a binary logistic regression classifier on embeddings from version $t_0$. We then evaluate the these classifiers on embeddings from version $t$ of the movies in the test set. Figure 2 (*right*) presents the area under the ROC curve averaged over the 19 genres (macro ROC-AUC). Focusing on the two post-processing methods, we observe that orthogonal alignment outperforms linear alignment. Interestingly, the three methods that modify the training procedure outperform *a*) both post-processing methods and *b*) classifiers retrained on embeddings of version $t$, indicating that the inductive biases introduced by these approaches can improve embedding quality beyond the alignment problem itself—a subtle point that is beyond the scope of our work.

## 4.2 Combining different models for text retrieval

Next, we consider a text retrieval application in which documents and queries are embedded with different models. This scenario arises when document embeddings are fixed and cannot be recomputed, e.g., because the raw documents are unavailable (Morris et al., 2023; Huang et al., 2024), but the query embedding model can be updated. Our main question is: Can retrieval performance be improved by upgrading the query embedding model, provided that embeddings are aligned?

We evaluate on three tasks from the retrieval subset of the MMTEB benchmark (Enevoldsen et al., 2025), summarized in Table 2 in Appendix B.2. Each of the three datasets (HotpotQA-HN, FEVER-HN, and TREC-COVID) consists of a corpus of text documents and a set of queries with ground-truth relevance labels. For each query, documents are ranked by the dot product between query and document embeddings. Performance is measured with the normalized discounted cumulative gain of the top-ten retrieved documents (nDCG@10).

We consider seven text embedding models publicly available on HuggingFace[2], varying in number of parameters, dimensionality, training objective, and release date. Several models are trained with Matryoshka representation learning (Kusupati et al., 2022), which enables truncation of embeddings at test time to trade accuracy for computational cost. Figure 3 (*left*) visualizes the models using the first two principal coordinates of the pairwise Procrustes distance matrix, computed on FEVER-HN document embeddings. Figure 3 (*right*) plots normalized Procrustes distance against dot-product preservation across all 21 model pairs. Empirically, the distances remain well below our theoretical worst-case bound and appear to approximately follow the power-law trend suggested by theory.

For each ordered pair of models, we learn an orthogonal transformation $Q^\star$ by sampling $10\,000$ documents uniformly at random from the corpus and solving the orthogonal Procrustes problem (1). When models have different dimensionalities, we pad the smaller embeddings with zeros, thus preserving their original geometry. We then embed all documents with the first model and all

---

[2]See: https://huggingface.co/models?pipeline_tag=sentence-similarity.

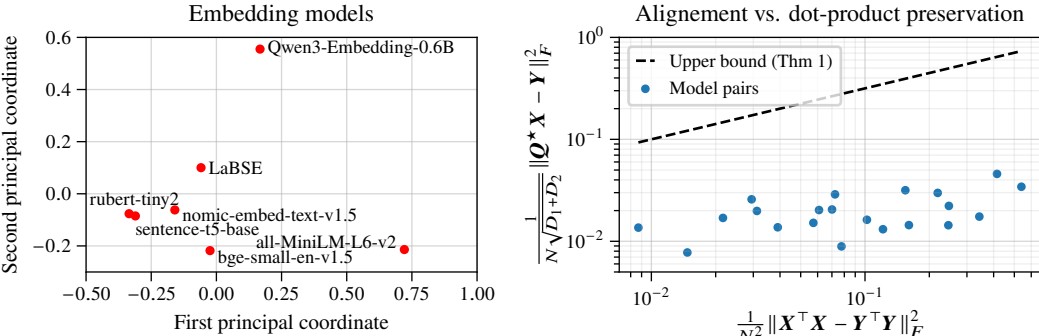

Figure 3: Two-dimensional representation of the text embedding models reflecting approximate Procrustes distances (*left*). Normalized Procrustes distance vs. geometry-preservation for all 21 pairwise model combinations (*right*).

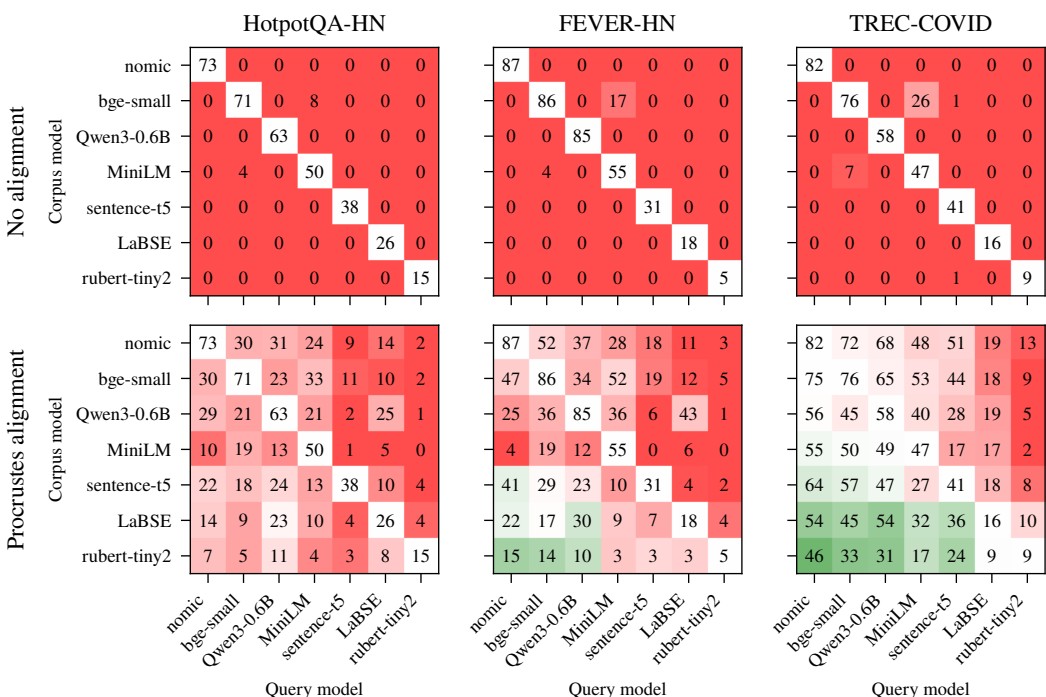

Figure 4: Retrieval performance (nDCG@10) for all query–document model combinations. *Top:* raw embeddings. *Bottom:* query embeddings aligned with orthogonal Procrustes. Diagonal entries correspond to the baseline case where the same model is used for both queries and documents.

queries with the second model, and evaluate retrieval performance in two settings, *a)* using raw query embeddings (no alignment), and *b)* aligning query embeddings with $Q^\star$ before retrieval. Figure 4 reports nDCG@10 for all 49 model pairs on the three tasks. Note that models are arranged in decreasing order of baseline performance. Without alignment, cross-model retrieval fails almost completely. After alignment, retrieval becomes feasible across models, and in two of the three tasks, upgrading to a stronger query model can yield substantial performance gains. In particular, the lower triangles in Figure 4 (*bottom*) show that replacing a weak query encoder with a stronger one, while keeping document embeddings fixed, can sometimes dramatically improve retrieval performance.

**Does the orthogonality constraint help?** We compare Procrustes alignment with an unconstrained linear alignment matrix $A^\star$ that minimizes the Frobenius error without enforcing orthogonality. By construction, the unconstrained solution cannot perform worse in terms of alignment error, as

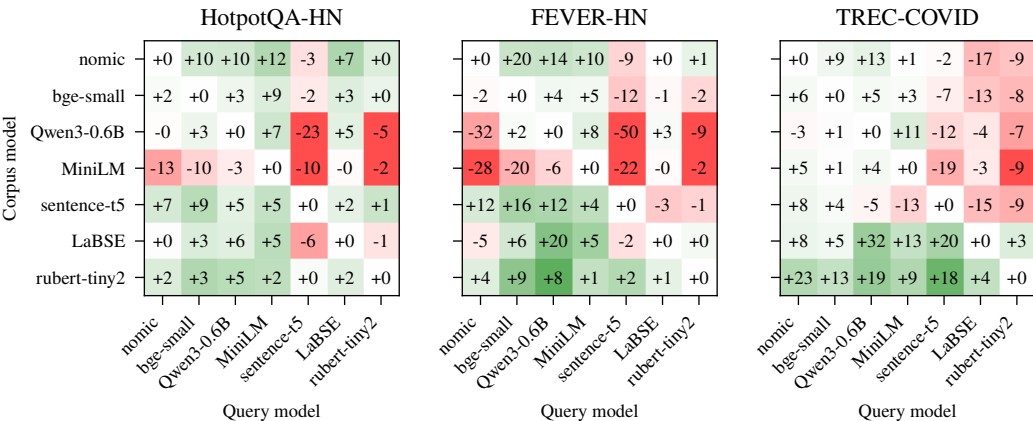

Figure 5: Difference in nDCG@10 between orthogonal Procrustes and unconstrained linear alignment. Positive values indicate orthogonal Procrustes performs better.

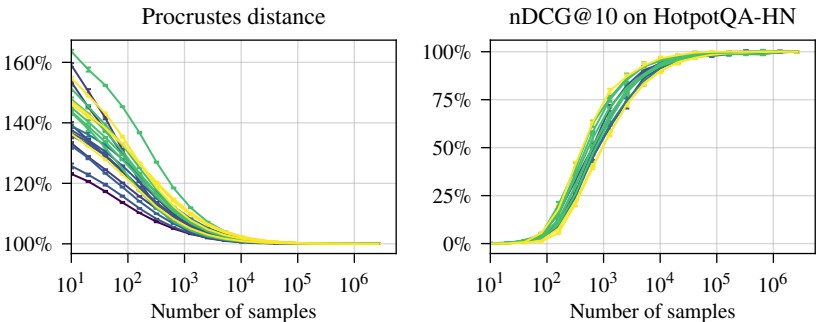

Figure 6: Performance vs. number of samples used to estimate $Q^\star$ across 21 model pairs, normalized by full-sample performance on HotpotQA. Brighter colors indicate more free parameters in $Q^\star$.

$\min_{A \in \mathbf{R}^{D \times D}} \|AX - Y\|_F \leqslant \min_{Q \in \mathcal{O}} \|QX - Y\|_F$. However, as shown in Figure 5, orthogonal alignment consistently outperforms linear, especially when upgrading to a stronger query model. This suggests that preserving the geometry of the stronger source model retains useful information that would otherwise be lost by unconstrained linear alignment. Conversely, when downgrading to a weaker query model (upper triangles), unconstrained alignment can help, but this case is less realistic.

**How many samples are needed to learn $Q^\star$?**    In order to learn the alignment matrix, we require a sample of texts embedded with both source and target embedding models. Figure 6 shows Procrustes distance and retrieval performance as a function of the number of training samples. For the models we consider, performance gains appear to saturate after roughly 10 000 samples, indicating relatively modest sample requirements for reliable alignment.

In Appendix B.2, we further analyze alignment matrices between models trained with Matryoshka representation learning (MRL). MRL encourages representations in which the leading dimensions capture most of the semantic variability. Consistent with this property, we find that $Q^\star$ between two Matryoshka models typically aligns the first 16–32 dimensions of one embedding space with the corresponding leading dimensions of the other.

### 4.3 IMPROVING MIXED-MODALITY SEARCH

Lastly, we consider an application of Procrustes post-processing to multimodal embedding models. Models such as CLIP and SigLIP train text and image encoders into a shared embedding space, enabling cross-modal retrieval (Wang et al., 2016). This allows, e.g., retrieving the most relevant images given a text query via dot-product comparisons, as in Section 4.2. Unlike the previous applications, the text and image encoders are jointly trained and therefore nominally aligned. However,

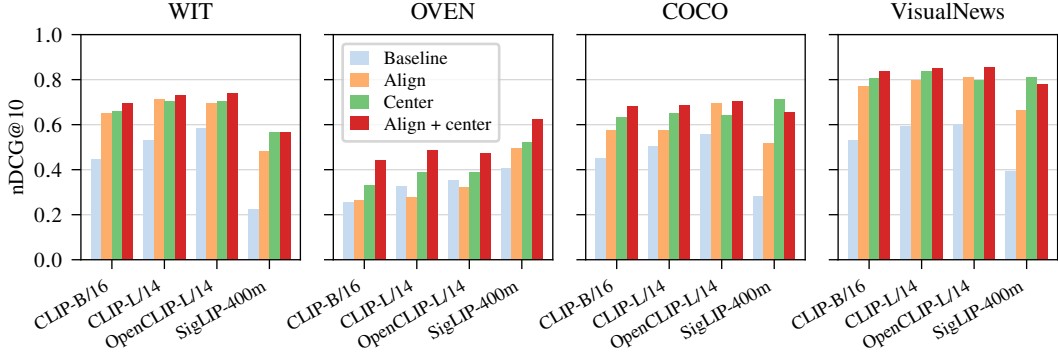

Figure 7: Retrieval performance (nDCG@10) on the four MixBench subsets. We evaluate four multimodal embedding models under different post-processing methods.

these models exhibit a persistent *modality gap*: embeddings cluster by modality in disjoint regions of $\mathbf{R}^D$ (Liang et al., 2022). This modality gap hinders comparisons between heterogeneous modalities, such as ranking a text-only and an image-only document with respect to a text query. This setting is known as mixed-modality search.

To systematically evaluate retrieval performance in this setting, Li et al. (2025) introduced the MixBench benchmark, building on four well-known multimodal text–image datasets. For each query (text in most subsets, or an image and a question in the OVEN subset), the goal is to retrieve the most relevant documents, which can be *a*) image-only, *b*) text-only, or *c*) an image–text pair. Embeddings for queries or documents that combine image and text are formed as weighted combinations: $x = \alpha x_{\text{text}} + (1 - \alpha) x_{\text{image}}$, where $\alpha$ is a hyperparameter. In their work, Li et al. demonstrate that simply centering and renormalizing the underlying text and image embeddings significantly improves retrieval, establishing the state of the art on MixBench. Note that centering modifies the dot-product distributions but does not explicitly align modalities (c.f. Appendix B.3). We hypothesize that explicitly aligning the embeddings of different modalities with orthogonal Procrustes can further improve the performance. We thus compare four variants, *a*) baseline (original unprocessed embeddings), *b*) orthogonal alignment only, *c*) centering only, and *d*) orthogonal alignment followed by centering.

In Figure 7, we report results on four multimodal embedding models. In these experiments, mean embeddings and alignment matrices are learned on held-out data derived from MixBench's upstream datasets. For mixed-modality queries and documents, we use $\alpha = 0.5$; results for a range of other values of $\alpha$ (presented in Appendix B.3) confirm the same qualitative trends. Across all subsets of MixBench and nearly all models, Procrustes post-processing improves mixed-modality retrieval. Orthogonal alignment alone outperforms the original unprocessed embeddings, while the combination of alignment and centering yields the best overall performance, consistently outperforming centering alone.

## 5    CONCLUSION & FUTURE WORK

We have shown that approximate dot-product preservation implies that two embedding models can be closely aligned by an orthogonal transformation, providing a principled justification for Procrustes alignment. Beyond this theoretical insight, we have demonstrated that Procrustes post-processing effectively addresses several practical challenges, including model retraining, partial upgrades, and multimodal search. These results highlight the growing importance of embedding alignment as machine learning systems increasingly interact in complex pipelines.

In future work, we plan to investigate alignment across modalities more deeply. Liang et al. (2022) show that the modality gap exists even at random initialization; we hypothesize that aligning representations at the start of training could improve optimization. More generally, we envision developing an alignment layer, similarly to normalization layers (Ioffe & Szegedy, 2015; Zhang & Sennrich, 2019), to make embedding interoperability a standard component of representation learning.

ETHICS STATEMENT

We have reviewed the ICLR code of ethics and we believe our work abides by it. Our work does not involve research or use of human subjects, and no potentially dangerous artifact is released as part of it. We study a general methodology for aligning representations that does not target any one particular application domain. We do not identify any potential societal implications (positive or negative) that can arise specifically as a consequence of this work.

REPRODUCIBILITY STATEMENT

All the models and datasets we use are publicly available. Upon publication, we commit to release code that will enable to reproduce all the experimental results presented in this paper. This will include data preprocessing, training and evaluation of our proposed methods, and any other methodologies we benchmark against.

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

# A PROOFS AND ADDITIONAL THEORY

In Section A.1, we provide a complete proof of Theorem 1. In Section A.2, we provide a concrete example of a pair of embedding matrices that achieves the upper bound.

## A.1 PROOF OF THEOREM 1

The Frobenius norm is a special case of the Schatten $p$-norm, which we will also make use of. Let $\boldsymbol{A} = [a_{ij}]$ be a real-valued matrix of rank $r$, with non-zero singular values $\sigma_1(\boldsymbol{A}) \geqslant \cdots \geqslant \sigma_r(\boldsymbol{A})$. For $p \in [1, \infty)$, the Schatten $p$-norm is defined as

$$\|\boldsymbol{A}\|_p = \left(\sum_{i=1}^r \sigma_i(\boldsymbol{A})^p\right)^{1/p}.$$

For $p = \infty$, it is defined as $\|\boldsymbol{A}\|_\infty = \sigma_1(\boldsymbol{A})$, in which case it coincides with the operator norm. We recover the Frobenius norm by setting $p = 2$.

In order to prove our bound, we will rely on a result first obtained by Powers & Størmer (1970, Lemma 4.1) in the case $p = 1$, and later extended to any $p$ by Kittaneh (1986, Corollary 2).

**Lemma 1** (Powers-Størmers-Kittaneh Inequality)**.** *Let $\boldsymbol{A}, \boldsymbol{B} \in \mathbf{R}^{N \times N}$ be positive semi-definite. Then,*

$$\|\boldsymbol{A} - \boldsymbol{B}\|_{2p}^2 \leqslant \|\boldsymbol{A}^2 - \boldsymbol{B}^2\|_p.$$

We also need another lemma that shows that the optimal alignment matrix aligns the subspaces spanned by the columns of $\boldsymbol{A}$ and $\boldsymbol{B}$.

**Lemma 2.** *Let $\boldsymbol{A}, \boldsymbol{B} \in \mathbf{R}^{M \times N}$ be such that $\mathrm{rank}(\boldsymbol{A}) \leqslant R$ and $\mathrm{rank}(\boldsymbol{B}) \leqslant R$. There exists an orthogonal matrix $\boldsymbol{P} \in \arg\min_{\boldsymbol{Q} \in \mathcal{O}} \|\boldsymbol{Q}\boldsymbol{A} - \boldsymbol{B}\|_F$ such that*

$$\mathrm{rank}(\boldsymbol{P}\boldsymbol{A} - \boldsymbol{B}) \leqslant R.$$

*Proof.* Let $\boldsymbol{U}\boldsymbol{\Sigma}\boldsymbol{V}^\top$ be a singular value decomposition of $\boldsymbol{B}\boldsymbol{A}^\top$ into $M \times M$ orthogonal matrices $\boldsymbol{U}, \boldsymbol{V}$ and an $M \times M$ diagonal matrix $\boldsymbol{\Sigma}$ containing the singular values sorted by magnitude, from largest to smallest. Schönemann (1966) shows that, for any such decomposition, the orthogonal matrix $\boldsymbol{P} \doteq \boldsymbol{U}\boldsymbol{V}^\top$ satisfies $\boldsymbol{P} \in \arg\min_{\boldsymbol{Q} \in \mathcal{O}} \|\boldsymbol{Q}\boldsymbol{A} - \boldsymbol{B}\|_F$. We know that $r \doteq \mathrm{rank}(\boldsymbol{B}\boldsymbol{A}^\top) \leqslant R \leqslant M$. If $r < M$, the last $M - r$ elements of the diagonal of $\boldsymbol{\Sigma}$ are zero, and $\boldsymbol{U}\boldsymbol{V}^\top$ is not unique. We will show that there is at least one pair $\boldsymbol{U}, \boldsymbol{V}$ that satisfies the claim.

Let $\mathrm{span}(\mathcal{S})$ be the linear subspace spanned by a set of vectors $\mathcal{S}$. For a matrix $\boldsymbol{M}$, let $\mathrm{col}(\boldsymbol{M})$ be the linear subspace spanned by its columns, and $\mathrm{null}(\boldsymbol{M})$ be its (right) nullspace. Assume that $\mathrm{rank}(\boldsymbol{A}) \leqslant \mathrm{rank}(\boldsymbol{B})$ and, without loss of generality, that $\mathrm{rank}(\boldsymbol{B}) = R$. By properties of the singular value decomposition and of the column space of matrix products, we have that

$$\mathrm{span}(\{\boldsymbol{u}_1, \ldots, \boldsymbol{u}_r\}) = \mathrm{col}(\boldsymbol{B}\boldsymbol{A}^\top) \subseteq \mathrm{col}(\boldsymbol{B}),$$

We can thus choose columns $r + 1, \ldots, R$ of $\boldsymbol{U}$ such that $\mathrm{span}(\{\boldsymbol{u}_1, \ldots, \boldsymbol{u}_R\}) = \mathrm{col}(\boldsymbol{B})$. Similarly, we know that

$$\mathrm{span}(\{\boldsymbol{v}_{r+1}, \ldots, \boldsymbol{v}_M\}) = \mathrm{null}(\boldsymbol{B}\boldsymbol{A}^\top) \supseteq \mathrm{null}(\boldsymbol{A}^\top),$$

and we can choose $\boldsymbol{V}$ such that $\mathrm{span}(\{\boldsymbol{v}_{R+1}, \ldots, \boldsymbol{v}_M\}) \subseteq \mathrm{null}(\boldsymbol{A}^\top)$. It follows that the last $M - R$ rows of $\boldsymbol{V}^\top \boldsymbol{A}$ contain all zeros. Letting $\boldsymbol{P} \doteq \boldsymbol{U}\boldsymbol{V}^\top$, we have that $\mathrm{col}(\boldsymbol{P}\boldsymbol{A}) = \mathrm{col}(\boldsymbol{U}\boldsymbol{V}^\top\boldsymbol{A}) \subseteq \mathrm{span}(\{\boldsymbol{u}_1, \ldots, \boldsymbol{u}_R\}) = \mathrm{col}\,\boldsymbol{B}$ by construction. In turn, we have that $\mathrm{col}(\boldsymbol{P}\boldsymbol{A} - \boldsymbol{B}) \subseteq \mathrm{col}(\boldsymbol{B})$, and we conclude that $\mathrm{rank}(\boldsymbol{P}\boldsymbol{A} - \boldsymbol{B}) \leqslant \mathrm{rank}(\boldsymbol{B})$.

If $\mathrm{rank}(\boldsymbol{A}) > \mathrm{rank}(\boldsymbol{B})$, we can swap the matrices $\boldsymbol{A}$ and $\boldsymbol{B}$ in the argument above and find $\boldsymbol{G} \in \arg\min_{\boldsymbol{Q} \in \mathcal{O}} \|\boldsymbol{Q}\boldsymbol{B} - \boldsymbol{A}\|_F$ such that $\mathrm{rank}(\boldsymbol{G}\boldsymbol{B} - \boldsymbol{A}) \leqslant R$. It is then easy to verify that setting $\boldsymbol{P} \doteq \boldsymbol{G}^\top$ verifies the claim. $\qquad\square$

It is interesting to note that this lemma holds for the Frobenius norm, but does not hold for all Schatten $p$-norms. We discuss this in more details in Appendix A.2.

Equipped with these, we can prove our main result.

*Proof of Theorem 1.* For a matrix $\boldsymbol{A} \in \mathbf{R}^{D \times N}$, define the matrix absolute value $|\boldsymbol{A}| = (\boldsymbol{A}^\top \boldsymbol{A})^{1/2}$ as the unique $N \times N$ positive semidefinite matrix such that $|\boldsymbol{A}|^\top |\boldsymbol{A}| = \boldsymbol{A}^\top \boldsymbol{A}$. The rank of $|\boldsymbol{A}|$ is equal to the rank of $\boldsymbol{A}$. We have that

$$\|\boldsymbol{X}^\top \boldsymbol{X} - \boldsymbol{Y}^\top \boldsymbol{Y}\|_F \geqslant \||\boldsymbol{X}| - |\boldsymbol{Y}|\|_4^2 \geqslant (2D)^{-1/2} \||\boldsymbol{X}| - |\boldsymbol{Y}|\|_F^2. \tag{2}$$

The first inequality follows from Lemma 1 with $p = 2$. The second inequality comes from the fact that, for any matrix $\boldsymbol{A}$ of rank $r$, with non-zero singular values $\sigma_1, \ldots, \sigma_r$,

$$\|\boldsymbol{A}\|_p = (\sigma_1^p + \cdots + \sigma_r^p)^{1/p} = (\mathbf{1}^\top \begin{bmatrix} \sigma_1^p & \cdots & \sigma_r^p \end{bmatrix})^{1/p}$$
$$\leqslant \left( \sqrt{r} \cdot \sqrt{\sigma_1^{2p} + \cdots + \sigma_r^{2p}} \right)^{1/p} = r^{1/2p} \|\boldsymbol{A}\|_{2p},$$

by the Cauchy-Schwarz inequality. In our case, $\boldsymbol{A} = |\boldsymbol{X}| - |\boldsymbol{Y}|$, and since $\mathrm{rank}(|\boldsymbol{X}|) \leqslant D$ and $\mathrm{rank}(|\boldsymbol{Y}|) \leqslant D$, the difference is of rank at most $2D$.

Furthermore, Lemma 2 states that there is an orthogonal matrix $\boldsymbol{G} \in \mathbf{R}^{N \times N}$ such that

$$\||\boldsymbol{G}|\boldsymbol{X}| - |\boldsymbol{Y}|\|_F \leqslant \||\boldsymbol{X}| - |\boldsymbol{Y}|\|_F \tag{3}$$

and $\mathrm{rank}(\boldsymbol{G}|\boldsymbol{X}| - |\boldsymbol{Y}|) \leqslant D$. This implies that there is an orthogonal matrix $\boldsymbol{H} \in \mathbf{R}^{N \times N}$ such that the last $N - D$ rows of $\boldsymbol{H}(\boldsymbol{G}|\boldsymbol{X}| - |\boldsymbol{Y}|)$ are all zeros. Let $\boldsymbol{S}, \boldsymbol{T} \in \mathbf{R}^{D \times N}$ be such that $\boldsymbol{S}$ coincides with the $D$ first rows of $\boldsymbol{HG}|\boldsymbol{X}|$, and $\boldsymbol{T}$ coincides with the $D$ first rows of $\boldsymbol{H}|\boldsymbol{Y}|$. By unitary invariance of the Frobenius norm and by construction of $\boldsymbol{H}$, we have that

$$\||\boldsymbol{G}|\boldsymbol{X}| - |\boldsymbol{Y}|\|_F = \|\boldsymbol{H}(\boldsymbol{G}|\boldsymbol{X}| - |\boldsymbol{Y}|)\|_F = \|\boldsymbol{S} - \boldsymbol{T}\|_F. \tag{4}$$

Since $\boldsymbol{S}^\top \boldsymbol{S} = \boldsymbol{X}^\top \boldsymbol{X}$, there is an orthogonal matrix $\boldsymbol{U} \in \mathbf{R}^{D \times D}$ such that $\boldsymbol{S} = \boldsymbol{UX}$. Similarly, there is an orthogonal matrix $\boldsymbol{V} \in \mathbf{R}^{D \times D}$ such that $\boldsymbol{T} = \boldsymbol{VY}$. By unitary invariance, we have that

$$\|\boldsymbol{S} - \boldsymbol{T}\|_F = \|\boldsymbol{UX} - \boldsymbol{VY}\|_F = \|\boldsymbol{PX} - \boldsymbol{Y}\|_F, \tag{5}$$

where $\boldsymbol{P} = \boldsymbol{V}^\top \boldsymbol{U}$. The claim follows by combining (2), (3), (4) and (5). $\square$

## A.2 TIGHTNESS OF UPPER BOUND

In this section, we provide an explicit example of a pair of embedding matrices that achieves equality in the upper-bound in Theorem 1. Let $D = 1$, $N = 2$, and let

$$\boldsymbol{X} = \begin{bmatrix} \sqrt{\frac{\varepsilon}{2\sqrt{2}}} & \sqrt{\frac{\varepsilon}{2\sqrt{2}}} \end{bmatrix}, \qquad\qquad \boldsymbol{Y} = \begin{bmatrix} \sqrt{\frac{\varepsilon}{2\sqrt{2}}} & -\sqrt{\frac{\varepsilon}{2\sqrt{2}}} \end{bmatrix}.$$

There are only two possible orthogonal transformations ($\pm [1]$), both of which align $\boldsymbol{X}$ and $\boldsymbol{Y}$ equally well. It is easy to verify that

$$\|\boldsymbol{X}^\top \boldsymbol{X} - \boldsymbol{Y}^\top \boldsymbol{Y}\|_F = \varepsilon, \qquad\qquad \max_{\boldsymbol{Q} \in \{\pm[1]\}} \|\boldsymbol{QX} - \boldsymbol{Y}\|_F = 2^{1/4} \sqrt{\varepsilon}.$$

This satisfies equality in the bound of Theorem 1. The example can be extended to $D > 1$ as follows. Let $\boldsymbol{e}_i \in \mathbf{R}^D$ be the $i$th standard basis vector, let $N = 2D$, and let $\boldsymbol{X}$ and $\boldsymbol{Y}$ be such that, for $i = 1, \ldots, D$,

$$\boldsymbol{x}_{2i-1} = \sqrt{\frac{\varepsilon}{2\sqrt{2}D}} \boldsymbol{e}_i, \qquad\qquad \boldsymbol{x}_{2i} = \sqrt{\frac{\varepsilon}{2\sqrt{2}D}} \boldsymbol{e}_i,$$

$$\boldsymbol{y}_{2i-1} = \sqrt{\frac{\varepsilon}{2\sqrt{2}D}} \boldsymbol{e}_i, \qquad\qquad \boldsymbol{y}_{2i} = -\sqrt{\frac{\varepsilon}{2\sqrt{2}D}} \boldsymbol{e}_i.$$

These embedding matrices also satisfy

$$\|\boldsymbol{X}^\top \boldsymbol{X} - \boldsymbol{Y}^\top \boldsymbol{Y}\|_F = \varepsilon, \qquad\qquad \max_{\boldsymbol{Q} \in \mathcal{O}_D} \|\boldsymbol{QX} - \boldsymbol{Y}\|_F = (2D)^{1/4} \sqrt{\varepsilon}.$$

For $D = 1$, the bound holds for all Schatten-$p$ norms, not only the Frobenius norm. However, for $D > 1$, this example can be used to show that the bound does not hold for general values of $p$.

## B ADDITIONAL EXPERIMENTAL DETAILS

This appendix mirrors the structure of the main text, with Section B.1 covering model retraining, Section B.2 covering partial upgrades, and Section B.3 covering multimodal embeddings.

Table 1: MovieLens experiment data

|  | **Partition1 (2019-02 - 07)** | **Partition2 (2019-03 - 08)** | **Partition3 (2019-04 - 09)** | **Partition4 (2019-05 - 10)** |
|---|---|---|---|---|
| **Partition1** | Ratings: 617,643 Users: 6,281 Movies: 9,537 |  |  |  |
| **Partition2** | Overlapping Users: 5,430 Movies: 8,949 | Ratings: 610,480 Users: 6,132 Movies: 9,452 |  |  |
| **Partition3** | Overlapping Users: 4,591 Movies: 8,501 | Overlapping Users: 5,258 Movies: 8,738 | Ratings: 610,296 Users: 6,091 Movies: 9,328 |  |
| **Partition4** | Overlapping Users: 3,944 Movies: 8,139 | Overlapping Users: 4,592 Movies: 8,320 | Overlapping Users: 5,391 Movies: 8,613 | Ratings: 594,011 Users: 6,007 Movies: 9,080 |

## B.1 MAINTAINING COMPATIBILITY ACROSS RETRAININGS

The experiments are conducted using the MovieLens-25M dataset, which contains 25 million ratings from $162\,541$ users on $59\,047$ movies between 2008 and 2019. We ignore the rating values and treat the ratings as binary implicit user feedback.

The core of our experiment involves training a matrix factorization-based BPR (Bayesian Personalization Ranking) model. This model is well-suited for implicit feedback, as it frames the learning process as a ranking task. During training, the model learns to rank an item the user has interacted with (i.e., a movie a user has rated) higher than an item the user is unlikely to have interacted with (i.e., a movie sampled uniformly at random from the set of movies the user has not rated). The model is optimized using the following objective function:

$$\ell_{\mathrm{BPR}}(\boldsymbol{V}, \boldsymbol{X}) = - \sum_{(u,i,j)\in D_s} \ln \sigma(\boldsymbol{v}_u \cdot \boldsymbol{x}_i - \boldsymbol{v}_u \cdot \boldsymbol{x}_j) + \lambda(\|\boldsymbol{V}\|_F^2 + \|\boldsymbol{X}\|_F^2)$$

Here, $u$ denotes a user, $i$ is a movie the user rated, and $j$ is a movie the user did not rate. $\boldsymbol{v}_u \in \mathbf{R}^D$ represents the learned embedding vector for user $u$, while $\boldsymbol{x}_i, \boldsymbol{x}_j \in \mathbf{R}^D$ represent the learned embedding vectors for movies $i$ and $j$, respectively. The equation represents the pairwise ranking loss, which seeks to maximize the difference between the positive and negative preferences.

The primary objective of this experiment is to evaluate the compatibility of embeddings across different training sessions. This phenomenon is particularly relevant in real-world scenarios where models are periodically retrained using new data. We simulate this industry practice by conducting multiple training runs on different time windows of the MovieLens-25M dataset.

Specifically, we create four distinct training partitions, each spanning a 6-month period. These partitions are sequentially aligned to simulate a rolling time window, with the data preceding four different months as the re-training time points: 2019-08 to 2019-11. For each partition, a standard preprocessing step is applied to ensure data quality. We filter the data to only include users and movies that have a minimum of 5 ratings within that specific partition. This preprocessing results in a different number of users, movies, and ratings in each partition, reflecting the natural evolution of the dataset over time. The counts for each partition and the overlapping between partitions are shown in table 1.

Hyperparameter tuning for the model is conducted using a separate, distinct dataset split. The training data for this process consists of 6 months of ratings between 2019-01 and 2019-06. Validation is performed on a subsequent 1-month period of data from 2019-07. The model is optimized using the Adam optimizer, with the number of training epochs fixed at 30. Hyperparameter tuning was performed using a grid search over the following parameter space.

- batch size: $\{512, 1024, 2048, 4096\}$
- embedding dimensionality: $\{8, 16, 32, 64\}$

- bias term for the movies: $\{\text{true}, \text{false}\}$
- learning rate: $\{1, 0.1, 0.01, 0.001\}$
- weight decay: $\{0.1, 0.01, 0.001, 0\}$

The validation task is a retrieval problem. For each user in the validation set, the model ranks all movies from the training data based on the dot product of the user and movie embeddings. The performance is measured by the Hit Rate at $K$ ($HR@K$), which quantifies whether a rated movie from the validation set appears within the top $K$ ranked movies for that user. Based on the performance on the validation set, measured by the Hit Rate at 100 ($HR@100$), the best configuration found was: batch size: 4096; embedding dimension size: 8; including movie bias: false; learning rate: 0.01; weight decay: 0.

In addition to the four partitions trained from scratch as baseline setting, we introduce three alternative training settings to explore methods for mitigating embedding drift and maintaining compatibility. These scenarios use the embeddings from the first partition (trained on data starting from 2019-02) as a reference point for the subsequent three partitions.

**Warmstart:** The training process for Partitions 2, 3, and 4 is initialized with the learned embeddings (weights) from Partition 1. The hyperparameters keep same as the baseline setting except the training epochs are decreased to 10.

**Autoencoding loss:** A regularization loss term is added to the training objective for Partitions 2, 3, and 4. This loss penalizes the distance between the newly learned embeddings and the embeddings from Partition 1 ($V_0, X_0$), encouraging them to stay close to the reference. The hyperparameters keep same as the baseline setting, and the regularization strength is set as $\lambda_{\text{auto}} = 1.0$.

$$\ell_{\text{auto}}(V, X) = \ell_{\text{BPR}} + \lambda_{\text{auto}}(\|V - V_0\|_F^2 + \|X - X_0\|_F^2)$$

**BC-Aligner:** This method introduces a learnable transformation matrix, $A$, which is co-trained with the user and movie embeddings for Partitions 2, 3 and 4. A regularization loss is applied to minimize the distance between the transformed embeddings ($AV$ and $AX$) and the reference embeddings from Partition 1 ($V_0$ and $X_0$), thus explicitly aligning the new embedding space with the first one. The hyperparameters keep same as the baseline setting, and the regularization strength is set as $\lambda_{\text{BC}} = 1.0$.

$$\ell_{\text{BC}}(V, X) = \ell_{\text{BPR}} + \lambda_{\text{BC}}(\|AV - V_0\|_F^2 + \|AX - X_0\|_F^2)$$

For the movie genre classification task, we use the movie metadata information in the MovieLens-25M dataset. It includes a genre list for each movie. The genres are selected from a list of 19 different genre terms.

## B.2 COMBINING DIFFERENT MODELS FOR TEXT RETRIEVAL

Table 2 introduces the three text retrieval tasks evaluated in Section 4.2, as well as two larger datasets used to sample training data to learn alignment matrices. Table 3 provides summary statistics for the text embedding models used in the experiments of that section. Figure 8 replicates the sample complexity analysis of Section 4.2 on the FEVER dataset. Qualitatively, the conclusions do not differ from those obtained on HotpotQA.

Figure 9 visualizes three alignment matrices, contrasting matrices that align two embeddings trained with MRL with matrices that align embeddings not trained with MRL. MRL encourages representations in which the leading dimensions capture most of the semantic variability. Consistent with this property, we find that $Q^\star$ between two Matryoshka models typically aligns the first 16–32 dimensions of one embedding space with the corresponding leading dimensions of the other.

## B.3 IMPROVING MIXED-MODALITY SEARCH

This section provides additional details pertaining to Section 4.3 in the main text. We start by arguing why centering is not necessarily a principled way to align different embedding spaces. Then, we provide information on our experimental setup as well as additional results.

Table 2: Summary statistics for the text retrieval datasets studied in Section 4.2. All datasets are part of the MMTEB benchmark (Enevoldsen et al., 2025).

| Name | # queries | # documents | Reference |
|------|-----------|-------------|-----------|
| HotpotQA-HN | 1000 | 225 621 | Yang et al. (2018) |
| FEVER-HN | 1000 | 163 698 | Thorne et al. (2018) |
| TREC-COVID | 50 | 171 332 | Roberts et al. (2021) |
| HotpotQA | — | 5 233 329 | Yang et al. (2018) |
| FEVER | — | 5 416 568 | Thorne et al. (2018) |

Table 3: Summary statistics of text embedding models used in the experiments of Section 4.2.

| Name | $D$ | Release date | Resizeable | Reference |
|------|-----|--------------|------------|-----------|
| nomic-embed-text-v1.5 | 768 | 2024-02 | Yes | Nussbaum et al. (2025) |
| bge-small-en-v1.5 | 384 | 2023-09 | No | Xiao et al. (2024) |
| Qwen3-Embedding-0.6B | 1024 | 2025-06 | Yes | Zhang et al. (2025) |
| all-MiniLM-L6-v2 | 384 | 2021-08 | No | N/A |
| sentence-t5-base | 768 | 2021-08 | No | Ni et al. (2022) |
| LaBSE | 768 | 2020-07 | No | Feng et al. (2022) |
| rubert-tiny2 | 312 | 2021-10 | No | N/A |
| bge-base-en-v1.5 | 768 | 2023-09 | No | Xiao et al. (2024) |
| gte-base-en-v1.5 | 768 | 2024-04 | Yes | Li et al. (2023) |

**Centering does not imply alignment.** Through an explicit example in two dimensions, we argue that centering embedding spaces does not necessarily help aligning them. Let

$$\boldsymbol{x}_1 = \begin{bmatrix} 1 \\ -\varepsilon \end{bmatrix}, \qquad \boldsymbol{x}_2 = \begin{bmatrix} 1 \\ +\varepsilon \end{bmatrix}, \qquad \boldsymbol{y}_1 = \begin{bmatrix} -\varepsilon \\ 1 \end{bmatrix}, \qquad \boldsymbol{y}_2 = \begin{bmatrix} +\varepsilon \\ 1 \end{bmatrix}.$$

Letting $\boldsymbol{\mu}_x = (\boldsymbol{x}_1 + \boldsymbol{x}_2)/2$ and $\boldsymbol{\mu}_y = (\boldsymbol{y}_1 + \boldsymbol{y}_2)/2$, and denoting the centered embeddings by $\tilde{\boldsymbol{x}}_i = \boldsymbol{x}_i - \boldsymbol{\mu}_x$ and $\tilde{\boldsymbol{y}}_i = \boldsymbol{y}_i - \boldsymbol{\mu}_y$, we have that

$$\tilde{\boldsymbol{x}}_1 = \begin{bmatrix} 0 \\ -\varepsilon \end{bmatrix}, \qquad \tilde{\boldsymbol{x}}_2 = \begin{bmatrix} 0 \\ +\varepsilon \end{bmatrix}, \qquad \tilde{\boldsymbol{y}}_1 = \begin{bmatrix} -\varepsilon \\ 0 \end{bmatrix}, \qquad \tilde{\boldsymbol{y}}_2 = \begin{bmatrix} +\varepsilon \\ 0 \end{bmatrix}.$$

Clearly, $\tilde{\boldsymbol{X}}$ and $\tilde{\boldsymbol{Y}}$ are not aligned ($\tilde{\boldsymbol{X}}^\top \tilde{\boldsymbol{Y}} = \boldsymbol{0}_{2\times 2}$), and arguably they are less aligned than the original embeddings $\boldsymbol{X}$ and $\boldsymbol{Y}$. On the other hand, observe that the orthogonal matrix

$$\boldsymbol{Q}^\star = \begin{bmatrix} 0 & 1 \\ 1 & 0 \end{bmatrix}$$

perfectly aligns the embeddings: $\bar{\boldsymbol{X}} \doteq \boldsymbol{Q}^\star \boldsymbol{X} = \boldsymbol{Y}$.

**Description of the models.** Table 4 provides a brief description of the different multimodal models we consider.

**Detailed experimental results.** Figure 10 presents retrieval performance for the four methods we consider as a function of the fusion weight $\alpha$. We observe that while the choice of $\alpha$ does impact absolute performance, the relative performance of different methods is relatively stable across a wide range of values.

### B.4 DATASETS AND TRAINING DETAILS

The MixBench benchmark (Li et al., 2025) builds on four large multimodal text-image datasets, *a*) Google WIT (Srinivasan et al., 2021), *b*) OVEN (Hu et al., 2023), *c*) COCO (Lin et al., 2014),

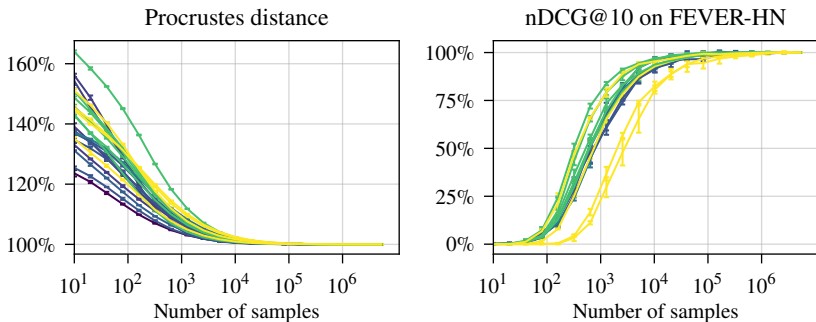

Figure 8: Performance vs. number of samples used to estimate $Q^\star$ across 21 model pairs, normalized by full-sample performance on FEVER. Brighter colors indicate more free parameters in $Q^\star$.

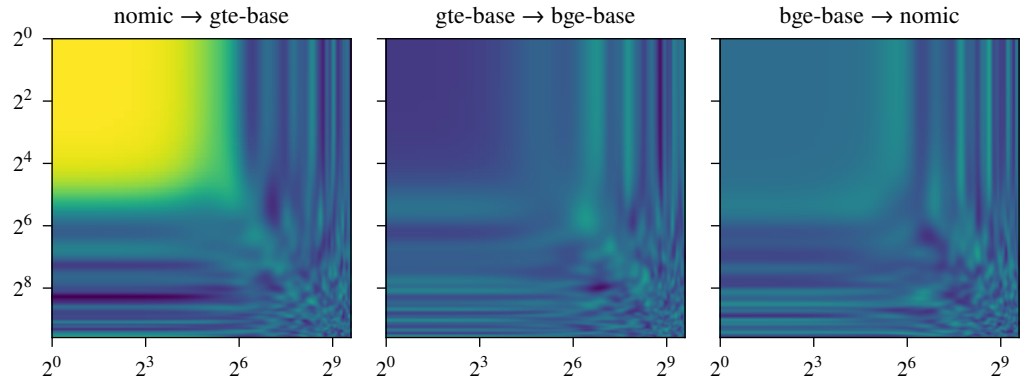

Figure 9: Visualization of orthogonal matrices aligning pairs of models. The matrix aligning nomic to gte-base tends to align the first 16–32 dimensions of nomic embeddings with the corresponding leading dimensions of gte-base embeddings.

and *d*) VisualNews (Liu et al., 2021). MixBench includes approximately 1000 multimodal query-document pairs extracted from these datasets. Details about the preprocessing steps used to obtain the final benchmark are provided in (Li et al., 2025, Appendix E).

To avoid estimating the cross-modality alignment matrix directly on the test data, we adopt the following procedure. We replicate the MixBench preprocessing pipeline and apply it to the training split of each of the four upstream datasets. From each dataset, we extract up to $10\,000$ unique text–image pairs, resulting in approximately $40\,000$ pairs in total. Given a multimodal model, we compute embeddings for all pairs and then fit a single alignment matrix using the orthogonal Procrustes method. This matrix is subsequently used to produce that model's experimental results on all four MixBench subsets. Our procedure closely follows (Li et al., 2025, Appendix B).

### B.5 ADDITIONAL PLOTS FOR REBUTTAL

Figures 11, 12, and 13 were added during the discussion phase.

## C LLM USAGE

We have used LLMs as general-purpose assisting tools for grammar, spelling and word choice in our manuscript, as well as for support with implementing code. No use of LLMs was made outside of these assistive purposes.

Table 4: Multimodal embedding models used for the experiments on MixBench.

| Name | $D$ | URL |
|------|-----|-----|
| CLIP-B/16 | 768 | `https://huggingface.co/openai/clip-vit-base-patch16` |
| CLIP-L/14 | 768 | `https://huggingface.co/openai/clip-vit-large-patch14` |
| OpenCLIP-L/14 | 768 | `https://huggingface.co/laion/CLIP-ViT-L-14-laion2B-s32B-b82K` |
| SigLIP-400m | 768 | `https://huggingface.co/google/siglip-so400m-patch14-384` |

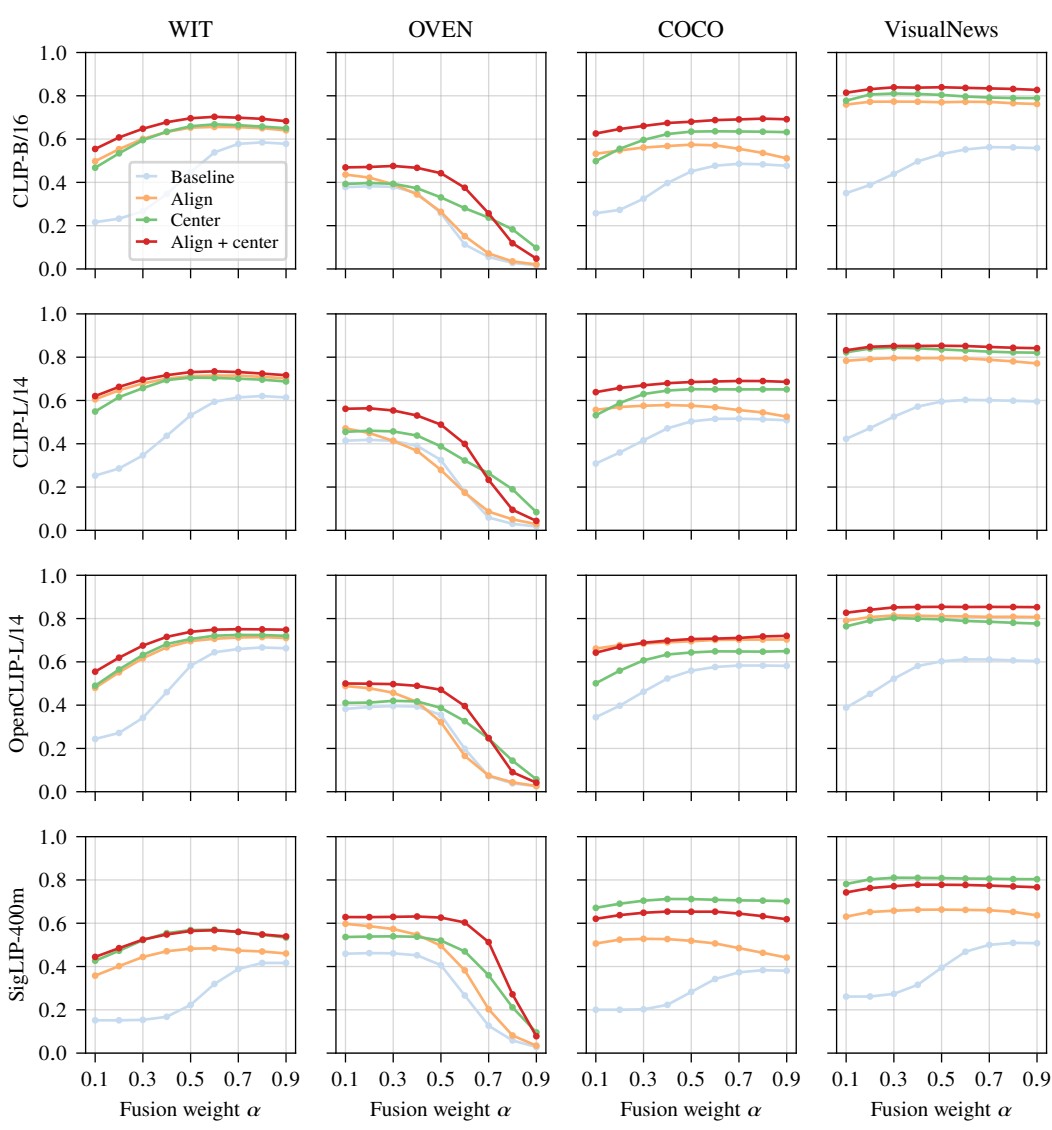

Figure 10: Retrieval performance (nDCG@10) on the four MixBench subsets, as a function of the fusion weight $\alpha$. We evaluate four multimodal embedding models under different post-processing methods.

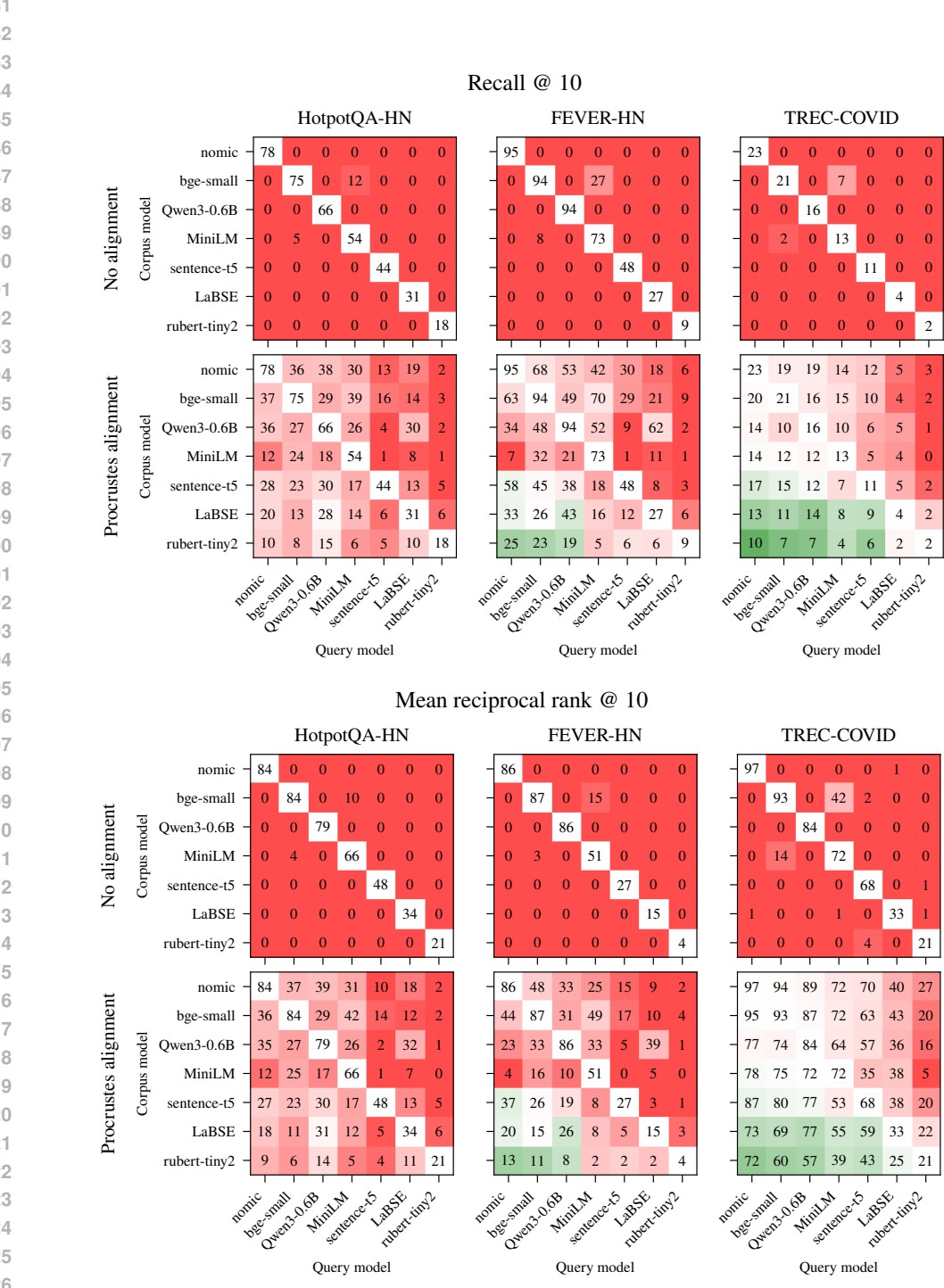

Figure 11: Retrieval performance (**recall @ 10 and MRR @ 10**) for all query–document model combinations. For presentation purposes, numbers are multiplied by 100. *Top rows:* raw embeddings. *Bottom rows:* query embeddings aligned with orthogonal Procrustes. Diagonal entries correspond to the baseline case where the same model is used for both queries and documents.

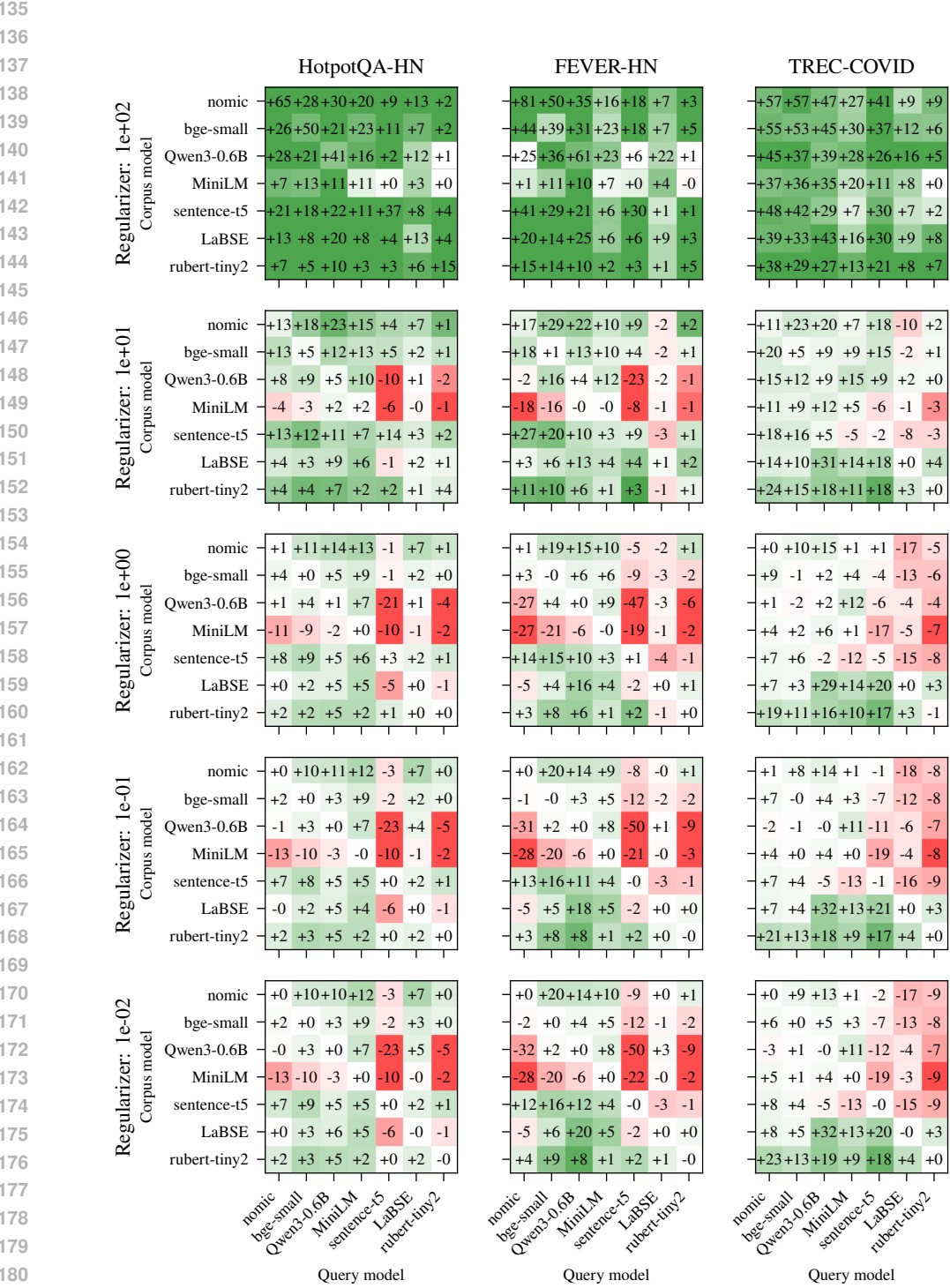

Figure 12: Difference in nDCG@10 between orthogonal Procrustes and unconstrained, **regularized** linear alignment. Rows correspond to different levels of $\ell_2$ regularization on the linear alignment matrix. Positive values indicate orthogonal Procrustes performs better. **No regularization (or very low regularization) on the linear alignment matrix works best.**

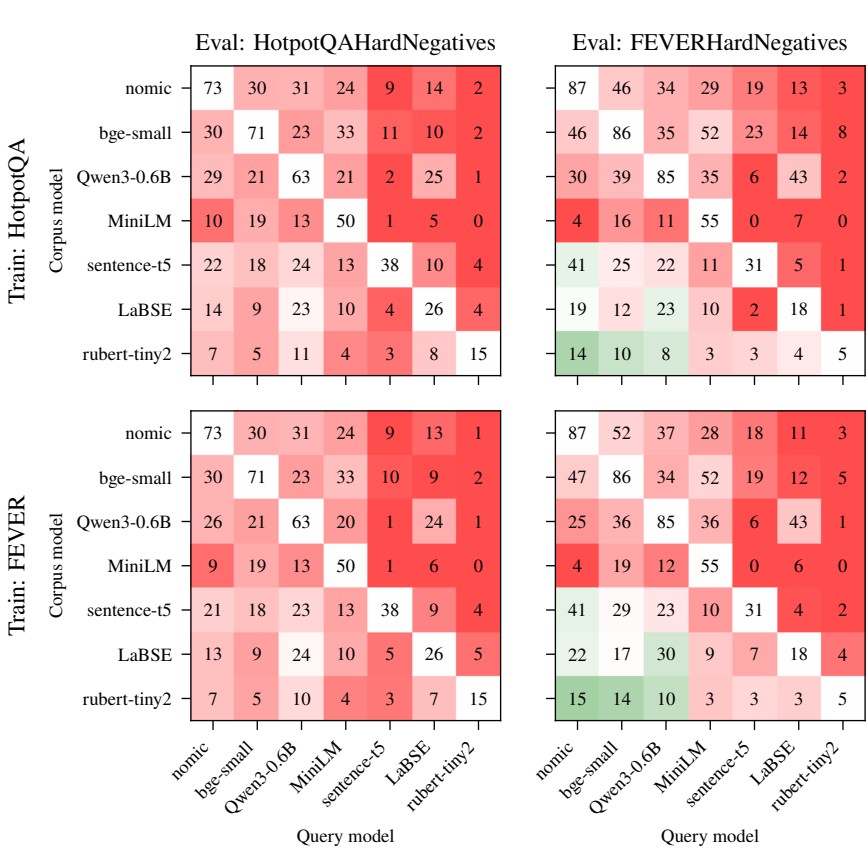

Figure 13: Retrieval performance (nDCG@10) for all query–document model combinations, when query embeddings are aligned with orthogonal Procrustes. Each row corresponds to a dataset that is used to estimate the orthogonal alignment matrix. Each column corresponds to a task that we evaluate on. **Take-away: alignment matrices learned on HotpotQA generalize well to FEVER and vice-versa.**

