# OpenReview forum: "When Embeddings Models Meet: Procrustes Bounds and Applications"
_ICLR.cc/2026/Conference — Submitted to ICLR 2026_

### Official Review · Reviewer_frDf · 2025-10-27

**Soundness:** 3
**Presentation:** 3
**Contribution:** 3
**Rating:** 6
**Confidence:** 4

**Summary:**

This paper investigates the fundamental problem of interoperability between separately trained embedding models, where similar semantic information is encoded but the representations are not directly interchangeable. The authors establish a strong theoretical foundation by proving a Procrustes Bound, which rigorously demonstrates that if the pairwise dot products (cosine similarities) between two embedding sets are approximately preserved, then a close-fitting orthogonal transformation (isometry) exists to align the spaces. This tight bound formally links similarity preservation to the existence of an effective linear alignment function. Leveraging this theoretical insight, the paper conducts a series of experiments validating Procrustes post-processing across 3 different embedding alignment use cases.

**Strengths:**

1. The paper provides a rigorous, novel Procrustes Bound which theoretically guarantees that if cosine similarities between two embedding sets are preserved, a close-fitting orthogonal transformation (isometry) exists. This provides a formal, tight linkage between embedding similarity and direct alignability.

2. The paper provides a thorough empirical analysis of post-alignment downstream performance after aligning embedding models across a variety of use cases.

**Weaknesses:**

1. While the derivation of the Procrustes Bound is mathematically rigorous, the result that dot-product preservation implies the existence of a linear alignment may be an expected consequence of standard contrastive or similarity-based training objectives used for all modern embedding models. Since these models are explicitly trained to make the dot product the central measure of semantic similarity, showing that a linear, orthonormal transformation aligns them when similarities are close is arguably a finding that validates the training objective, rather than an independent, deep insight into the structure of representation space. The paper could be strengthened by expanding their analysis to non-similarity-preserving embedding spaces.

**Questions:**

N/A

---

> ### Author Response · Authors · 2025-11-19
>
> Thank you for your review and for your supportive comments!
>
> > The paper could be strengthened by expanding their analysis to non-similarity-preserving embedding spaces.
>
> You are correctly pointing out that the starting point of our work is loss functions that are orthogonally invariant, e.g. methods that define similarity through dot-products. We are less well-versed in the area of non-similarity-preserving embeddings spaces and would welcome your suggestions for additional analyses (e.g. concrete models, tasks or applications) you believe could strengthen our work.

---

### Official Review · Reviewer_A5Ld · 2025-10-31

**Soundness:** 3
**Presentation:** 3
**Contribution:** 3
**Rating:** 6
**Confidence:** 3

**Summary:**

The paper tackles the task of aligning embeddings produced by different models since embedding models trained separately on similar data often produce representations that encode stable information but are not directly interchangeable. It proposes a method that uses orthogonal Procrustes transformations and prove that if two embedding spaces approximately preserve pairwise dot products, there exists an orthogonal transformation that closely aligns them. Finally, the authors test the proposed approach in several scenarios: maintaining compatibility across retrainings, combining different models for text retrieval, and improving mixed-modality search.

**Strengths:**

The paper tackles an important problem. It provides both a theoretical foundation and also shows empirically that the proposed method achieves good results. I feel like this idea can be very useful for the research community.

**Weaknesses:**

My main concern is related to the comparison with state of the art which I find relatively shallow. While probably not directly comparable, it would be good to understand the difference in performance to other alignment methods, some of them being based on the Procrustes transformations such as:

[Quantized Wasserstein Procrustes Alignment of Word Embedding Spaces](https://aclanthology.org/2022.amta-research.15/) (Aboagye et al., AMTA 2022)

This leads me to my next point, about a somehow limited novelty since the idea of using the orthogonal Procrustes transformations for embedding alignment was studied before as the authors point out so it would be good to better emphasize the differences with existing methods.

**Questions:**

* What are some failure and limitations of the proposed method?
* Does the method handle embeddings of different dimensionalities beyond zero-padding? What are the effects of zero-padding vs performance?

---

> ### Author Response · Authors · 2025-11-19
>
> Thank you for your thoughtful and constructive review which highlights several important points.
>
> > My main concern is related to the comparison with state of the art which I find relatively shallow. While probably not directly comparable, it would be good to understand the difference in performance to other alignment methods, some of them being based on the Procrustes transformations such as [ref].
>
> Thank you for bringing that paper to our attention. That work addresses a more challenging problem, similar to the one we discuss in lines 157-158 of related work section (Grave et al., 2019). In contrast to our work, theirs does not assume access to pairs of embeddings that relate to the same object. Concretely, whereas we assume that $\boldsymbol{x}_i$ relates to the same object as $\boldsymbol{y}_i$, their work assumes there are no such “matching subscripts” (so-to-speak).
>
> The procedure they develop alternates between steps where embeddings are matched, and steps where the matched embeddings are aligned. Our work only addresses the second part. The _alignment_ part of their approach is identical to ours; it is also finding and applying the orthogonal Procrustes transformation.
>
> To address your concern about comparisons and baselines more broadly, let us briefly describe how we think about alignment methods more generally. We believe the fundamental feature that distinguishes alignment methods is whether geometry is preserved or not.
>
> - **Geometry-preserving**: orthogonal Procrustes is the only method. There is no other method we can compare to in this category.
> - **Others**: once we let go of geometry-preservation, alignment can be cast as a regression problem, and any regression model can be used. We pick one representative method here: linear regression.
>
> The trade-off is preserving the information in the embedding (geometry-preservation) versus optimizing alignment. In the absence of established alternative methods from the literature, we believe that comparing orthogonal Procrustes to linear regression is sufficient to rigorously explore this trade-off.
>
> The exception is successive versions of embeddings for recommendations (Sec 4.1), where there are competing methods in the literature (El-Kishky et al, 2022, Hu et al, 2022). We include these in our experiments.
>
> > This leads me to my next point, about a somehow limited novelty since the idea of using the orthogonal Procrustes transformations for embedding alignment was studied before as the authors point out so it would be good to better emphasize the differences with existing methods.
>
> Thank you for raising this important point, You are very much correct that orthogonal Procrustes is a well-known idea. Let us clarify our contribution.
>
> - Our main contribution is a theoretical result that relates geometry-preservation to the error after Procrustes alignment (i.e., Procrustes distance). This is original and significant: while orthogonal Procrustes has long been used as a heuristic, our result provides a principled justification, shedding light on _why_ & _when_ it works.
> - Our experimental results rigorously demonstrate the benefits of Procrustes alignment on several applications, including ones where orthogonal Procrustes has not been applied to previously (mixed-modality search). To some extent, this is less original, as you point out. Nevertheless, it enables us to reach new SOTA results on mixed-modality search (sec. 4.3).
>
> > What are some failure and limitations of the proposed method?
>
> Our work argues that orthogonal Procrustes has many attractive properties and is a very strong method empirically across a broad range of applications, as far as alignment methods go.
>
> However, our results also show that performance can sometimes degrade when combining different models, even after alignment. This is highlighted, e.g., in the results of Figure 4. In short: alignment methods are not a silver bullet (whether orthogonal Procrustes or otherwise), and there can be a price to pay when combining different embedding models.
>
> > Does the method handle embeddings of different dimensionalities beyond zero-padding? What are the effects of zero-padding vs performance?
>
> Orthogonal Procrustes can indeed align embeddings of different dimensionalities - it does not require embeddings to have the same dimensionality. Zero-padding is a useful way to reason about the “different dimensionality” setting, but in practice it is not necessary to actually zero-pad the embeddings. We can simply learn an alignment matrix of dimensions $D_1 \times D_2$ using the same simple SVD-based algorithm.
>
> In particular, there is no specific impact on performance (whether alignment error or computational cost) of handling embeddings of different dimensionalities.

---

### Official Review · Reviewer_5xpT · 2025-11-04

**Soundness:** 3
**Presentation:** 3
**Contribution:** 2
**Rating:** 2
**Confidence:** 4

**Summary:**

This paper studies when two sets of embeddings can be aligned using orthogonal transformations (Procrustes alignment). The authors provide a tight theoretical bound showing that if dot products are approximately preserved between embedding sets, then there exists an orthogonal transformation that can align them well. They demonstrate this approach across three applications: maintaining compatibility during model retraining, combining different models for text retrieval, and improving mixed-modality search.

**Strengths:**

Well-motivated: The paper addresses a practically important challenge, efficiently aligning embedding spaces without costly re-indexing, which is highly relevant for production systems.
Comprehensive experiments: The evaluation covers three diverse applications with thorough experimental design, including baselines and ablations.
Clear presentation: The paper is well-written with good intuitive explanations and effective visualizations.

**Weaknesses:**

My main concern with this paper is the limited empirical impact: The key assumption that embedding models preserve dot products often doesn't hold in practice as model updates typically involve shifts in embedding geometry (otherwise there would be no need to update them). This is also mirrored by performance limitations: The results show that these simple transformations (whether orthogonal or not) rarely achieve performance between the source and target models as desired. Strong improvements only occur when the source model is very weak, limiting practical utility: in reality the updates are more likely to be incremental wrt performance - large gains can offset the cost of re-indexing.

Overall, while the paper addresses an important problem with solid theory and thorough experiments, the practical impact is limited due to restrictive assumptions and modest empirical gains in realistic settings.

**Questions:**

No questions, the paper is better suited for a more focused venue.

---

> ### Author Response · Authors · 2025-11-19
>
> We appreciate your careful assessment and understand that some elements of our contribution may not have been sufficiently communicated. We provide clarifications below.
>
> > The key assumption that embedding models preserve dot products often doesn't hold in practice as model updates typically involve shifts in embedding geometry (otherwise there would be no need to update them).
>
> We think there is a fundamental misunderstanding here. Let us take the text embedding models and the retrieval tasks of Section 4.2 as an example. If the geometry wasn’t approximately the same across models, then it would be impossible to have several models simultaneously achieve non-trivial performance on the retrieval tasks. (In a way, these retrieval tasks are simply testing the embedding models’ geometry.)
>
> Very concretely, we hope you agree with the following: any two reasonable text encoders will generate embeddings for “cat”, “kitten” and “encyclopedia” such that dot(“cat”, “kitten”) > dot(“cat”, “encyclopedia”). Of course different models will be slightly different and there is space for newer models to better capture semantic similarity, but effectively the differences are small relative to the overall similarity in geometry across models.
>
> Also, note that our theory doesn’t make any strict assumption on dot-product preservation. All our theory says is: “how well Procrustes aligns two sets of embeddings depends only on how well the dot-products are preserved”.
>
> > The results show that these simple transformations (whether orthogonal or not) rarely achieve performance between the source and target models as desired.
>
> What do you consider to be “performance as desired”?
>
> In Section 4.3, by using orthogonal Procrustes, we improve state-of-art mixed-modality search performance on 14 out of 16 experiments. In Section 4.2, we improve cross-model retrieval over the baseline in 124 out of 126 experiments we run.
>
> In Section 4.2, our experiments do indeed show that performance often degrades when combining two different models. If possible, it might be better to stick to using a single model in those cases. This is expected. In fact, it was a surprise to us that upgrading the query model does sometimes improve performance (we thought it would merely limit how much performance degrades).
>
> Our claim is as follows. _If you need to align two different embedding models, orthogonal Procrustes has appealing properties in theory, and in practice will get you very far_. If you can avoid having to combine and align embedding models, then you might be better off doing so. (But unfortunately that is not always possible).
>
> If you had specific expectations for what the method should achieve, we would be grateful for further detail so we can better address the concern.

---

### Official Review · Reviewer_6uk4 · 2025-11-05

**Soundness:** 2
**Presentation:** 3
**Contribution:** 1
**Rating:** 4
**Confidence:** 4

**Summary:**

The paper investigates the theoretical analysis Procrustes analysis. It establishes that if the pairwise dot products between the two embedding matrices are approximately preserved, then there is a tight upper bound on the Procrustes distance. The experiments show that procruste outperforms other baseline transformation on Model retraining, Partial upgrades and Multimodal embeddings.

**Strengths:**

1. The bound is mathematically tight and improves on earlier results.
2. Experiments are comprehensive and demonstrate practical relevance across diverse applications.

**Weaknesses:**

1. Unverified Assumption: The main theorem assumes that pairwise dot products between embeddings are approximately preserved. However, the paper does not empirically verify whether this assumption holds for real-world embedding models in the application section.
2. Lack of Theoretical–Practical Connection: While the theory focuses on bounding the alignment error (upper bound), the experiments mainly show that Procrustes performs well empirically. Are there any synthetic experiments that verify the upper bound’s dependence on $D^{1/4}$ and $\epsilon^{1/2}$.
3. No New Methodology or Improvement: The paper provides theoretical justification for the standard Procrustes method but does not propose a new algorithm or modification inspired by the theory. As a result, the contribution is primarily analytical rather than methodological.
4. Limited Applicability: The Procrustes method inherently requires embeddings to have the same dimensionality and assumes a orthogonal relationship. In real-world scenarios where embedding spaces differ in dimension or where the transformation is nonlinear, this approach is not directly applicable.

**Questions:**

1. Under what specific conditions does the Procrustes method work effectively, and in what situations would it fail ?
2. Why don’t the authors compare Procrustes with other commonly used alignment methods, such as Canonical Correlation Analysis (CCA) or Centered Kernel Alignment (CKA), which can also measure or enforce embedding similarity?

---

> ### Author Response · Authors · 2025-11-19
>
> Thank you for your thorough review. Your comments are very valuable in helping us clarify our claims and improving our presentation, and we are already planning to do several changes to the manuscript based on your review.
>
> We believe we can convincingly address many of the points you raise, and we have attempted to do so below. Let us know you have any remaining concerns or if you see other opportunities to improve the manuscript.
>
> > Unverified Assumption: The main theorem assumes that pairwise dot products between embeddings are approximately preserved. However, the paper does not empirically verify whether this assumption holds for real-world embedding models in the application section.
>
> Our main theorem doesn’t make any assumption about dot-product preservation, it simply _relates_ dot-product preservation to error after alignment (i.e., Procrustes distance). One can always set $\varepsilon = \lVert \boldsymbol{X}^\top \boldsymbol{X} - \boldsymbol{Y}^\top \boldsymbol{Y} \rVert$ in Theorem 1. There is no restrictive assumption there.
>
> We will reformulate Theorem 1, changing “assume that $\lVert \boldsymbol{X}^\top \boldsymbol{X} - \boldsymbol{Y}^\top \boldsymbol{Y} \rVert \le \varepsilon$” to “let $\varepsilon = \lVert \boldsymbol{X}^\top \boldsymbol{X} - \boldsymbol{Y}^\top \boldsymbol{Y} \rVert$”. Thanks for pointing this out and giving us an opportunity to clarify.
>
> Perhaps what your question is also getting at is the following: what is a $\varepsilon$ that is “small enough” for the alignment to be “successful”? We believe there is no general answer; it is an empirical question that depends on the model and on the downstream task. In some sense, all our experiments implicitly attempt to address this question.
>
> Taking the text retrieval experiments to illustrate this point. Clearly, text embedding models do preserve dot products to a very large extent (or at least the ranking of dot products), otherwise it would be impossible for many of these models to achieve non-zero scores on the retrieval tasks. Our experiments show (implicitly, through non-trivial performance on a downstream task), that they can be aligned well.
>
> > Lack of Theoretical–Practical Connection: While the theory focuses on bounding the alignment error (upper bound), the experiments mainly show that Procrustes performs well empirically. Are there any synthetic experiments that verify the upper bound’s dependence on D^(¼) and epsilon^(½).
>
> As we state in lines 187-188, the bound is tight, and the dependence in $D$ and $\varepsilon$ is optimal. We do better than a synthetic experiment: in Appendix A.2, we provide a specific construction, with explicit embedding matrices $\boldsymbol{X}$ and $\boldsymbol{Y}$ parametrized by $D$ and $\varepsilon$, where equality is achieved in Theorem 1 (we show this analytically).
>
> We have reworked Appendix A.2 in the latest version of the PDF and we hope the latest version completely clarifies this point.
>
> > Limited Applicability: The Procrustes method inherently requires embeddings to have the same dimensionality and assumes a orthogonal relationship. In real-world scenarios where embedding spaces differ in dimension or where the transformation is nonlinear, this approach is not directly applicable.
>
> The Procrustes method **does not** require the embeddings to have the same dimensionality. For example, the text embedding models we mix & match in Section 4.2 have different values of $D$ ranging from 312 to 1024 (see Table 3).
>
> Conceptually, to align two sets of embeddings of different dimensionalities (say, $D_1 < D_2$), one can simply zero-pad the “smaller” embeddings to match the size of the “larger” ones (clearly, zero-padding preserves dot products in the smaller embeddings). When going from large to small, simply drop the $D_2 - D_1$ trailing dimensions after applying the alignment matrix. In practice, zero padding is not even necessary and one can learn a $D_1 \times D_2$ (or $D_2 \times D_1$) alignment matrix directly.
>
> Furthermore, our approach absolutely works for nonlinear transformations. In fact, our work is _motivated_ by nonlinear transformations. Our theory shows that there is always a linear (orthogonal) transformation that aligns two embeddings models, with an alignment error that depends only on how well dot-products are preserved. This is the case no matter how nonlinear the underlying transformation between the two embedding matrices is.
>
> We plan to explicitly discuss these two points in the next version of the manuscript, given some additional space.

---

> > ### Author Response · Authors · 2025-11-19
> >
> > > Under what specific conditions does the Procrustes method work effectively, and in what situations would it fail ?
> >
> > The informal take-away from our theoretical result is the following: “how well Procrustes works depends only on how well the dot-products are preserved”. We want to stress that this is a non-trivial insight because, as you point out above, Procrustes is a linear method, and the two embedding models might be highly non-linear (e.g. different encoder architectures, etc).
> >
> > Again, your question might be getting at: in practice, how well should the dot-products be preserved for downstream performance of aligned embeddings to be good enough? As discussed above, this depends on the models, on the task, on performance requirements, etc. The empirical part of our paper shows that, in several realistic and general real-world use cases, Procrustes alignment can be very effective.
> >
> > > Why don’t the authors compare Procrustes with other commonly used alignment methods, such as Canonical Correlation Analysis (CCA) or Centered Kernel Alignment (CKA), which can also measure or enforce embedding similarity?
> >
> > CCA and CKA can measure similarity between embeddings, but they do not explicitly provide a way to align one set of embeddings with another.
> >
> > The alignment procedure that is conceptually most similar to CCA is multivariate linear regression of $\boldsymbol{Y}$ on $\boldsymbol{X}$ (see [Kornblith et al, 2019](https://arxiv.org/pdf/1905.00414)). We compare Procrustes alignment with linear regression in Sections 4.1 and 4.2. Our high-level take-away is that preserving geometry (a feature of Procrustes alignment) usually performs better.

---

### Official Review · Reviewer_2Zdm · 2025-11-11

**Soundness:** 2
**Presentation:** 2
**Contribution:** 2
**Rating:** 4
**Confidence:** 4

**Summary:**

The authors study when two independently trained embedding spaces that encode stable information but are not directly interchangeable can be made interoperable via an orthogonal procrustes alignment. The main theoretical result in the paper is that if pairwise dot products are approximately preserved, i,e. The gram matrices are close in frobenius norm, then there exists an orthogonal transformation with procrustes bounded error. The algorithmic alignment uses standard orthogonal procrustes solved via SVD. Empirical results are shown on three applications, model retraining with retrieval and classification, partial upgrades for text retrieval and mixed modality search.

**Strengths:**

1. The theoretical bound and results are simple and improves prior dependence on N in settings which have large datasets.

2. Orthogonal procrustes is computationally simple and it keeps the source space geometry. So the overall idea is practical and relevant for the community.


3. The experiments cover different settings such as retraining, cross-model retrieval and multimodal search, which is exhaustive as far as the applications are concerned.

**Weaknesses:**

1. There are mismatches between the theory and practice. The experiments vary the number of training pairs but there is no formal generalization bound relating gram-matrix closeness on a sample to the alignment error on the full data. Also the theory requires pairwise dot product and same indexed objects for a paired X, Y in the SVD computation step, however for setting mentioned in the partial updates where raw documents are unavailable, where would the paired data come from and how representative are the measurements then?

2. Some of the experimental details are unclear. For multimodal, it is not mentioned which modality is rotated and how the pairs are sampled. For retrieval, no MRR or Recall@k is reported. The results on classification with retraining interestingly show that training time methods beat post hoc alignment including retraining, however the authors don’t discuss this.


3. The paper is missing some analysis and experiments on orthogonal with scale and orthogonal with whitening, and regularized linear baselines.

**Questions:**

See Weaknesses section above.

---

> ### Author Response · Authors · 2025-11-19
>
> We appreciate your thoughtful and constructive review. Your feedback will help us refine the manuscript. We are confident we can address many of your concerns, as detailed below.
>
> > There are mismatches between the theory and practice. The experiments vary the number of training pairs but there is no formal generalization bound relating gram-matrix closeness on a sample to the alignment error on the full data.
>
> What you describe would require relating the alignment error on a sample to the alignment error on the full data (The first step - relating gram-matrix closeness on a sample to alignment error on the same sample - follows from our Thm 1). In the ICLR 2024 paper [Estimating Shape Distances on Neural Representations with Limited Samples](https://openreview.net/forum?id=kvByNnMERu), Pospisil et al. study this problem. Their Theorem 1 addresses your question partly, but requires additional assumptions on the embeddings. In short, this is a challenging problem.
>
> We do what we believe is the next best thing: a rigorous empirical investigation of the effect on using limited samples on the Procrustes distance and on the downstream performance of the aligned vectors (see Fig. 6 and 8).
>
> Note that there are important applications where the full set of vectors is known, and generalization is not an issue. For example, aligning successive versions of user and item embeddings (Sec 4.1).
>
> Taking a step back, we think the key insight our theory provides is very practical: approximately preserving dot-products means embeddings can be closely aligned. Based on our experience working in the field, this is non-obvious to many practitioners. Arguably, the fact that alignment matrices sometimes need to be estimated from limited samples and need to generalize to new embeddings is easier to grasp.
>
> > Also the theory requires pairwise dot product and same indexed objects for a paired X, Y in the SVD computation step, however for setting mentioned in the partial updates where raw documents are unavailable, where would the paired data come from and how representative are the measurements then?
>
> In practice, it is sufficient to have pairs that are coming from a reasonably similar distribution. For example, if the documents are unavailable, we can learn the alignment matrix on a different set of documents (processsing each document with both embedding models to construct a paired set).
>
> We have added experiments in Appendix B.5 to validate this claim. Figure 13 shows that estimating the alignment matrix on a subset of HotpotQA and evaluating the performance on FEVER works almost as well as estimating the matrix on FEVER directly (and vice-versa).
>
> > Some of the experimental details are unclear. For multimodal, it is not mentioned which modality is rotated and how the pairs are sampled.
>
> - **Which modality is rotated**: Because the absolute position of the embeddings is irrelevant, it does not matter which modality is considered “source” and which is considered “target” in the alignment procedure - both choices yield the same alignment, in terms of relative position. In our code, we treat images as the target modality and texts as the source, but all the results would be rigorously identical if we had done it the other way around.
> - **How the pairs are sampled**: we have added a section (Appendix B.4) detailing how the alignment matrix is trained. We obtain text-image pairs from the training splits of four multimodal datasets (WIT, OVEN, COCO and VisualNews), similarly to how the MixBench dataset is constructed. We apologize for the oversight, and we hope that the new section in the Appendix fully addresses your question.
>
> (Note that even though we evaluate on each subset of MixBench separately, we use a single alignment matrix trained from a mix of data. This is another small piece of evidence in favor of the claim that alignment matrices are relatively robust to the choice of training data.)

---

> > ### Author Response · Authors · 2025-11-19
> >
> > > The results on classification with retraining interestingly show that training time methods beat post hoc alignment including retraining, however the authors don’t discuss this.
> >
> > We discuss it briefly in lines 294-297. Our experiments do indeed suggest that, in the context of this task, tweaks to the training procedure can improve performance over the standard training recipe we use.
> >
> > The subtle point here is that the benefits of orthogonal Procrustes are distinct from any specific training recipe. The key practical benefit of orthogonal Procrustes is geometry-preservation. It decouples the training recipe from the need to have aligned embeddings, enabling a clear separation of concerns.
> >
> > - When training a model, pick whichever training method maximizes model performance. Informally, “find the best geometry”.
> > - Then, use orthogonal Procrustes to optimally align the embeddings, without affecting the geometry. Informally: maximize alignment while staying fully “performance-neutral”.
> >
> > We could have applied Procrustes alignment on the embeddings produced by the alternative training methods, knowing with confidence that the resulting model would be at least as good, and might even be better.
> >
> > Given more space in the manuscript, we would gladly add this discussion in the paper.
> >
> > > The paper is missing some analysis and experiments on orthogonal with scale and orthogonal with whitening, and regularized linear baselines.
> >
> > In the context of the text retrieval experiments, we have added experimental results on regularized linear baselines in Figure 12 (Appendix B.5). In our experimental setup, where we learn the linear alignment matrix on 10k samples, regularization does not help (and eventually starts hurting performance).
> >
> > We believe that the two other baselines you suggest are not meaningful in our setting.
> >
> > - **Orthogonal with scale**: In Sections 4.2 and 4.3, all the models produce unit-norm embeddings by construction (such that the dot-product coincides with the cosine similarity). Furthermore, all the results we report, including those of Section 4.1, only depend on the relative ranking of dot-products, which is invariant to scale. In other words, orthogonal with scale would produce results that are exactly identical throughout the paper
> > - **Orthogonal with whitening**: the applications we are interested in pre-suppose that the existing (target) embedding space cannot be changed. Pre-processing the target embeddings (e.g. whitening them) is not possible.

---

### Author Response · Authors · 2025-11-19

We thank all the reviewers once again for their constructive feedback. Their comments helped us identify several points that required clarification and have contributed to improving the clarity and presentation of the paper. **We believe we have addressed the main concerns raised in the reviews, and we welcome any further questions on remaining issues**.

We have uploaded a revised PDF with the following changes:

- Minor rewording of Theorem 1 and Corollaries 1 & 2 to make it clear that there is no restrictive assumption on dot-product preservation
- Corrected a small typo in Corollary 2 that we noticed post-submission.
- Improved Appendix A.2, which provides an example that achieves equality in the bound of Theorem 1, highlighting the tight dependence on $D$ and $\varepsilon$.
- Details on how the alignment matrix for the multimodal experiment is trained in Appendix B.4.
- Additional experimental results for the text retrieval tasks in Appendix B.5:
    - Figure 11: alternative metrics: MRR @ 10, recall @ 10
    - Figure 12: regularized linear baselines
    - Figure 13: out-of-distribution generalization

---

### Author Response · Authors · 2025-12-03

### Summary of Paper Discussion and Our Responses

For convenience, we provide a brief summary of points raised by the reviewers, and how we addressed them.

**Theoretical assumptions not verified in practice.** (2Zdm, 6uk4)

Informally, our main result simply says: "how well Procrustes works depends only on how well dot-products are preserved". There is no restrictive assumption. The reviewers' valuable feedback led us to change the wording of our theorem and corollaries to emphasize this.

**Some of the experimental details are unclear.** (2Zdm)

We have provided additional clarification in our response to the reviewer (we plan to fold this into the mansucript). We have also added a new section in the PDF (Appendix B.4) with details on the mixed-modality search experiment.

**Missing baselines.** (2Zdm)

We have added experiments with regularized linear alignment matrices in the PDF. In our response to the reviewer, we explain why the other methods they mention (Procrustes with scale, whitening matrices beforehand) are not applicable to our setting.

**Applicability when documents are missing.** (2Zdm, A5Ld)

We have clarified that, in practice, we can learn the alignment matrix on any set of matched documents. We have added experimental results to the PDF analyzing out-of-distribution performance (where alignment matrices are trained on documents from one dataset and evaluated on documents from another, distinct dataset).

**Verifying the upper bound's dependence on $D$ and $\varepsilon$** (6uk4)

We have substantially improved Appendix A.2, which provides an explicit construction parametrized by $D$ and $\varepsilon$ that achieves equality in our bound. This demonstrates that our bound is tight.

**Limited Applicability / different dimensionalities** (6uk4, A5Ld)

We have clarified that two embedding spaces of different dimensionalities can be handled seamlessly. It is straightforward to extend our theory to that setting, and in fact our experiments already _do_ combine models of different dimensionalities.

**Comparison to CCA or CKA** (6uk4)

In our response, we highlight a connection between a special case of CCA and linear regression (which we do include as a baseline in most of our experiments). We also explain why, while CCA and CKA are useful similarity metrics, they cannot be used to align different embeddings.

**Assumptions of orthogonality** (6uk4, 5xpT)

We have clarified that there is no such assumption in our work. What we show is that, if similarity scores are approximately preserved across two embedding spaces (e.g., two models that capture the meaning of English text), then an orthogonal transformation is a good approximation.

**Limited novelty** (A5Ld)

In our response, we have further clarified how we view our contributions. In short, we believe that

1. our theoretical insight is a significant and novel contribution,
2. our application of Procrustes to mixed-modality search is novel and potentially impactful, and
3. the remaining experiments are less novel; but they provide a rigorous empirical perspective that complements our theory.

### Why We Think This Paper Should be Accepted.

We address an important and timely problem, i.e., efficient alignment of embeddings learned by different machine learning models. We do so by relating approximate preservation of dot products with distances between Procrustes-aligned embeddings. We believe that the ICLR community would benefit from understanding the properties of such Procrustes alignment, given its application to many problems of practical interest.

---

### Meta-Review · Area_Chair_t6cW · 2026-01-06

**Summary:**

This paper studies the orthogonal Procrustes problem for an orthogonal transforming a set of embeddings from another such that it minimizes the average squared distance between the two sets of embeddings. They propose a new bound on the error that is data-independent and depends only on the dimensionality and the difference in gram matrices or intuitively how closely dot products are preserved across each embedding. Prior works either had data-dependent bounds or had conditions such as centered embeddings. Their experiments investigates the effectiveness of orthogonal Procrustes across three setups: model retraining, partial upgrades, mixed-modality search.

The authors during the rebuttal highlight that their theory provides the following insight: “How well Procrustes aligns two sets of embeddings depends only on how well the dot-products are preserved”. The authors believe this is a key insight and practical as approximately preserving dot-products means embeddings can be closely aligned.

The reviewers note various strengths including a novel bound, thorough empirical analysis, tackling an important problem, clear presentation, and computationally simple.

**Reviewer Concerns:**

Reviewer frDf:
- **Extension to non-similarity-preserving embedding spaces**: The reviewer argues that the results may be expected because the paper only considers contrastive or similarity-based objectives that are based on preserving the dot product similarity. The authors acknowledge this requirement.

Reviewer A5Ld:
- **Comparison with state-of-the-art such as Aboagye et al., AMTA 2022**: The authors argue the reference does not assume access to pairs of embeddings relating to the same object and their alignment step is the same as theirs. They claim that orthogonal Procrustes is the only geometry-preserving method and for non-geometry preserving methods they consider a representative algorithm using a regression model and believe that this comparison is sufficient to rigorously explore the trade-off between geometry-preservation versus optimizing alignment.
- **Limited novelty**: The reviewer notes that orthogonal Procrustes is not a novel method and using it for the mentioned problem is also not novel. The authors note that their main contribution is a theoretical result and that orthogonal Procrustes has not been applied previously to mixed-modality search where they achieve state-of-the-art results (Section 4.3)
- **What are failures and limitations?** The authors note that the performance can sometimes degrade when combining different models even after alignment and refer to Figure 4.
- **Embeddings with different dimensionalities**: The method does support different dimensionalities even without zero-padding.

Reviewer 5xpT:
- **Limited empirical impact and limiting key assumption**: The authors claim that “if the geometry was not approximately the same across models, then it would be impossible to have several models simultaneously achieve non-trival performance on the retrieval tasks.”. The AC finds this claim unclear and non-rigorous while crucial to the core of the paper. It is unclear why a global measure of geometry-preservation would be needed for retrieval while only local geometry to each point matters the most and as long as farther away points are far enough the global geometry does not matter.
- **Strong improvements only occur when the source model is very weak**: The authors refer to sections where the method helps improve state-of-the-art performance.

Reviewer 6uk4:
- **Unverified Assumption of dot product preservation**: The authors argue their theorem does not make any assumptions about dot-product preservation and relates the dot-product preservation to error after alignment.
- **Lack of Theoretical–Practical Connection and request for synthetic experiments to verify the upper bound**: The authors provide a synthetic experiment in Appendix A.2 where the equality is achieved analytically.
- **No New Methodology or Improvement and Limited Applicability**: The authors note that the method does not require the same dimensionalities and their approach works for nonlinear transformations.
- **Specific conditions for effectiveness**: The authors state that “how well Procrustes works depends only on how well the dot-products are preserved” and emphasize that this is not a trivial insight but the effectiveness depends on the task requirements.
- **Comparison to other alignment methods such as CCA and CKA**: The authors note that CCA and CKA are primarily designed to measure similarity between embeddings and the alignment procedure most similar to CCA is multivariate linear regression. Because of that they argue comparison to linear regression is enough.

Reviewer 2Zdm:
- **No formal generalization bound relating Gram-matrix closeness to the alignment error**: The authors argue this is a challenging problem and they provide empirical investigation.
- **Setting with unavailable raw paired data**: The authors argue it is sufficient to have pairs from a reasonably similar distribution and provide new experiments in Appendix B.5.
- **Which modality is rotated and which pairs are sampled?** The authors note that it does not matter which modality is considered as the results will be the same. They provide more information in Appendix B.4 for how the alignment matrix is trained.
- **For retrieval, no MRR or Recall@k is reported.** The authors added Figure 11.
- **Missing experiments on orthogonal with scale/whitening and regularized linear baselines**: The authors added results on regularized linear baselines in Figure 12 (Appendix B.5). The authors believe that the other two baselines are not meaningful as orthogonal with scale would be identical to without and orthogonal with whitening is not applicable to their setup because they assume the target embeddings cannot be pre-processed. The AC notes that this argument is not convincing and unclear from the paper where this constraint is justified.

**Reviewer Scores:**

Reviewers gave scores of 2, 4, 4, 6, 6. The reviewers did not engage during the rebuttal and did not change their scores. The AC acknowledges that the proposed bound is tight and rigorous. However, the AC finds the limited novelty and contributions a shared concern among reviewers which has been noted in various ways. The authors note that their novelty is in providing a novel theoretical insight, novel application to mixed-modality search, and rigorous empirical perspectives. However, the AC believes providing a tight bound without connecting the bound directly to the experimental results is not a strong contribution. As it is right now, the experiments are somewhat detached from the theoretical contribution of the paper and do not have enough contribution by themselves.

There are various ways that the paper could be made more impactful as noted by reviewers:
- Comparison to more non-geometry-preserving methods (A5Ld).  The authors may consider expanding the comparison to methods that partially preserve geometry, for example, ones that preserve only the local geometry.
- Justifying why preserving the geometry matters and to what extent including relating Gram-matrix closeness to the alignment error (2Zdm).
- Novel empirical applications in problems where Procrustes has not been applied before.
- Extension to non-similarity-preserving embedding spaces (frDf). The authors may consider adding a categorization of common objective functions and rigorously defining what is considered similarity-preserving.

---

### Decision · Program_Chairs · 2026-01-26

Reject